



# Seasonal differences in observed versus modeled new particle formation over boreal regions

Carl Svenhag[1], Pontus Roldin[1,3], Tinja Olenius[2], Robin Wollesen de Jonge[1], Sara M. Blichner[4], Daniel Yazgi[2], and Moa K. Sporre[1]

[1]Department of Physics, Lund University, Lund, Sweden
[2]Swedish Meteorological and Hydrological Institute, Norrköping, Sweden
[3]Swedish Environmental Research Institute IVL, Malmö, Sweden
[4]Department of Environmental Science and Analytical Chemistry, Stockholm University, Stockholm, Sweden

**Correspondence:** Carl Svenhag (carl.svenhag@fysik.lu.se)

**Abstract.** Realistic representation of atmospheric aerosol size distribution dynamics in large scale climate models is important for developing accurate descriptions of aerosol-cloud interactions. Despite the dynamic nature of the distributions, which have large seasonal and diurnal changes, model evaluations often focus on the annual median size distribution. Using more comprehensive monthly and diurnal model illustrations can be crucial for evaluating model performance and potential aerosol effects for short term variations. In this study, we assess the impact of a molecular model scheme for $NH_3-H_2SO_4$ nucleation integrated into the Earth System Model (ESM) EC-Earth3, across different seasons, months, and days within the boreal climate during the year 2018. Measured number size distributions from two in-situ boreal stations are used to evaluate and to study particle formation and growth representation in EC-Earth3 over 2018. Additionally, we utilize results from the ADCHEM model, a state of the art 1-D Lagrangian aerosol-chemistry model. This allows us to compare EC-Earth3 against results from highly detailed model description of aerosol formation and growth at the boreal stations. When comparing diurnal EC-Earth3 model results with ADCHEM and observations, we establish that using solely organic-$H_2SO_4$ nucleation parameterization will underestimate the aerosol number concentrations. The new added $NH_3-H_2SO_4$ nucleation parameterization in this study improves the resulting aerosol number concentrations and reproduction of particle formation events with EC-Earth3. However, from March to October, the EC-Earth3 still underestimates particle formation and growth.

## 1 Introduction

Aerosol particles suspended in the atmosphere are constantly forming and growing around us. *Secondary* aerosols are particular matter formed in the atmosphere from precursor gases, either by the condensation of vapors onto existing particles or by the nucleation of new particles. Secondary aerosols have been shown to significantly impact the global climate and human health (Szopa et al., 2021; Nault et al., 2021; Pye et al., 2021). New-particle-formation (NPF) occurs when particles are formed from the clustering and condensation of various gases. The gases can originate from both natural and anthropogenic sources. A major gas compound driving new-particle formation is sulfuric acid ($H_2SO_4$). Atmospheric $H_2SO_4$ is formed in the gas-phase when $SO_2$ is oxidized by OH radicals, and when dimethyl sulfate (DMS) is oxidized by OH, Cl, and BrO radicals (see e.g.



Wollesen de Jonge et al. 2021). Sulfur dioxide is predominantly emitted from anthropogenic sources and volcanoes, while DMS is mainly emitted by phytoplankton in the surface ocean. Secondary aerosols involving organic compounds (SOA) are produced from volatile organic compounds (VOCs) in the atmosphere. 85% of the VOCs are estimated to originate from natural sources (Lamarque et al., 2010; Guenther et al., 2012), labelled biogenic volatile organic compounds (BVOCs). The abundance of these natural gas compounds can have strong fluctuations over the year, in relation to the growing season. Furthermore, gas-phase ammonia ($NH_3(g)$) has been shown to be a key species in the NPF processes by recent model and experimental studies (Dunne et al., 2016; Roldin et al., 2019). Atmospheric $NH_3$ is mainly a by-product emission from agricultural and industrial activities. Including $NH_3$ in the NPF parameterization has not yet been well established in most large-scale climate models, so evaluating the impact of the recently introduced $NH_3$-enhanced formation mechanism is a key focus of this study.

Quantification of aerosol formation mechanisms has advanced significantly in the past decades with the development of computational atmospheric chemistry models. The essential prerequisites for achieving reasonable modelling of aerosols and NPF is to define an accurate representation of the aerosols' physical and chemical processes. However, the price for increasing the complexity in models has to be balanced with the model's computational capacity. All models are restricted within limited computational process time. Generally, the complexity of model chemistry and physics is limited by the size of the spatial and temporal scale it has to cover. Therefore, it is more common for global Earth system models (ESMs) to have more crude parameterization for aerosols and chemistry compared to e.g. a 1-D model. Nonetheless, improving the representation of aerosols in Earth system models is crucial for reducing the large uncertainties associated with aerosols regarding global climate forcing (Forster et al., 2021).

Atmospheric aerosols tend to contribute to a net cooling of the global climate by scattering incoming short-wave solar radiation, here referred to as the direct radiative effect (DRE) given in units of $Wm^{-2}$. Higher aerosol concentration can also increase the number of cloud condensation nuclei (CCN), which can promote clouds to have higher cloud droplet number concentrations (CDNC) and cloud albedo. This will increase the cloud's ability to effectively scatter short-wave radiation, which promotes cooling of the climate as a consequence (Twomey, 1974; Albrecht, 1989). The changes in cloud radiative effect (CRE) from aerosols are also mainly promoting net negative (cooling) climate forcings. The aerosol-cloud effect is considered to produce a stronger net forcing than the direct scattering from aerosols (Forster et al., 2021). Nevertheless, aerosol-cloud-climate effects hold some of the highest uncertainty of our known climate forcers. It is worth mentioning that recent studies also suggests that increased NPF production can significantly decrease the CRE outcome. This can occur when a large portion of the available vapors are consumed by strong NPF events, which limits the availability of vapors for existing particles to grow into the CCN sizes (Sullivan et al., 2018; Roldin et al., 2019; Blichner et al., 2021; Patoulias et al., 2024).

Results from Svenhag et al. (2024) showed that the cloud radiative effect (CRE) in the Earth system model (ESM) EC-Earth3 exhibits high sensitivity to changes in the new particle formation scheme. Similarly, Sporre et al. (2020) showed the input SOA yields and VOC precursors in EC-Earth3-AerChem resulted in high sensitivity in the CRE. The Sporre et al. (2020) study also showed that different ESMs can produce opposite CRE outcomes from removing a VOC precursor in the models. This further implies that the modelled NPF and sub-100 nm aerosol dynamics have a strong effect on the CRE, and it can vary significantly between Earth system models. Furthermore, evaluation of ESMs and radiative effects commonly uses temporal averaging to



monthly mean or median outputs from the model (Mann et al., 2014; Sporre et al., 2020; Svenhag et al., 2024), and this could discount shorter extreme NPF events which could have a large impact on the CRE locally.

The current default NPF parameterization scheme in EC-Earth3-AerChem include two nucleation pathways from: $H_2SO_4$ with water (relative humidity), and $H_2SO_4$ with extremely low-volatile organic compounds (ELVOCs) (Vehkamäki, 2002; Riccobono et al., 2014; Bergman et al., 2022). Using our new molecular clustering model scheme, here referred to as CLUST, incorporated through lookup table in EC-Earth3-AerChem, we introduce the $H_2SO_4-NH_3$ NPF mechanism in the EC-Earth3 aerosol module M7. The look-up table uses $H_2SO_4$, $NH_3$, temperature, ion-production, and cluster scavenging sink to derive
the nucleation rate (Svenhag et al., 2024). A recent extensive study by Zhao et al. (2024) indicated the importance of including various NPF mechanisms in global models. They model and discuss how implementing 11 unique pathways had a significant effect on the vertical and horizontal aerosol size distributions.

A difficult task for all global model studies is to accurately validate the model, especially for aerosol formation and growth of sub-100 nm aerosols. This is partly due to the lack of available aerosol measurements globally. The global distribution
of measurement stations is scarce, aerosol sizes below 10 nm in diameter are rarely measured, and observations of vertical profiles of aerosol distributions are lacking. The coarse spatial resolution in the large-scale models also makes extrapolating a ground station measurement to a $2° \times 3°$ (latitude $\times$ longitude) grid-box likely non-representative. The modal aerosol size distribution representation in EC-Earth3, which is typical in ESMs, also hinders us to compare growth rates similar to what is seen in measured observations. Additionally, observations from satellite instrumentation can not confidently measure aerosols
in the sub-100 nm diameter sizes, and aerial campaign measurements for vertical distributions are not sufficient yet.

For this study, we selected to compare and validate our EC-Earth simulations against two rural boreal stations in Sweden and Finland. The grid-boxes in EC-Earth where the stations are based are rural with predominantly flat topography and are not heavily impacted by anthropogenic influences. This makes the extrapolation of the station observations to the model EC-Earth grid more accurate, as the areas are expected to share a more homogeneous air mass (Nieminen et al., 2014). At the Swedish
site, we would expect higher $NH_3$ and $H_2SO_4$ levels compared to the Finnish site due to its proximity to more agricultural and urban regions.

The focus of this study is to determine the model's ability to predict NPF events and to investigate the timing of these events in the model. In addition to comparing the results to observations, we also compare EC-Earth simulations to results from the detailed chemistry process model ADCHEM. ADCHEM is operated as a Lagrangian 1-D column model along air
mass trajectories, with similar nucleation and growth schemes as EC-Earth but including detailed aerosol size distribution dynamics and gas-phase and aerosol chemistry. This enables evaluation of the more simplified approaches used in ESMs against detailed modeling, which is essential for understanding the ESM performance. Comparisons to only field measurements of NPF do not give full information on the reasons for the differences between model and observations, for example, whether they originate from missing chemical components or from inadequate approximations used for included components. They
also do not guarantee that a model is right for the right reasons. Therefore, we apply ADCHEM results to benchmark and understand the aerosol formation representation in EC-Earth. This might reveal missing model features in EC-Earth3 that can be addressed in future development.



## 2 Method

### 2.1 EC-Earth3.4.0

The Earth system model configuration used in this study is the EC-Earth3-AerChem (version EC-Earth3.4.0). This contains the global circulation model Integrated Forecast System (IFS) cycle 36r4, which is coupled to the atmospheric chemistry model Tracer Model 5 (TM5) version 1.2 with the carbon bond 5 (CB05) setting (Krol et al., 2005; van Noije et al., 2014). EC-Earth3 uses the model coupler OASIS3-MCT version 3.0 (Craig et al., 2017) where the information exchange between IFS and TM5-MP is made every 6-hours, model time. The IFS model time step is 45 minutes and set to generate output every 3 hours on

a 0.7°spectral truncation grid. TM5 uses hourly time steps and is set to produce hourly model output with 2° × 3° (latitude × longitude) resolution. The model is run as atmosphere-only, where EC-Earth3 is fed fixed sea-surface temperatures (SST) and sea-ice content from the AMIP reader (van Noije et al., 2021). The vertical resolution in TM5 is represented by 34 hybrid sigma pressure levels, and IFS have the same hybrid pressure levels, but extrapolated to 91 layers.

#### 2.1.1 Aerosol module M7

The size-resolved aerosol micro-physics module M7 (Vignati et al., 2004), here used in TM5, has been developed for use in large-scale transport models (e.g. Earth system models). Application of M7 within ESMs is motivated by the modal-based size distribution that increase their computational efficiency. The size distribution in M7 is described by seven log-normal modes with fixed standard deviation. This way, the model solely has to track the aerosol mass and the aerosol number of each mode for every time-step. This is generally much more computationally efficient compared to a sectional scheme, which

requires significantly more tracers (Vignati et al., 2004). M7 uses four water-soluble (S) and three insoluble (I) size modes with restricted diameter ranges as: nucleation mode (NUS) for d < 10 nm, Aitken modes (AIS,AII) for 10 < d < 100 nm, accumulation (ASC,ASI) 100 < d < 1000 nm, and coarse modes (COS,COI) for d > 1000 nm. However, this modal system in M7 limits the size distribution appearance to specific dimensions, and the transfer of aerosol mass and number between modes have some assumptions which are considered unrealistic (Vignati et al., 2004). M7 includes seven aerosol species: sea salt

(SS), dust (DU), black carbon (BC), sulfate (SO$_4$), primary organic aerosol (POA), and secondary organic aerosols (SOA).

#### 2.1.2 New particle formation and growth

The formation of new particles in M7 is modeled by calculating a rate ($J$) of newly formed 5 nm diameter aerosols every time step in the units s$^{-1}$cm$^{-3}$. The default nucleation rate parameterization (Riccobono et al., 2014) in M7 used for EC-Earth3-AerChem is:

$$J_{Riccobono} = K_m \left[H_2SO_4\right]^2 \left[ELVOC\right] \tag{1}$$

The equation includes a constant empirical factor ($K_m = 3.27 \times 10^{-21}$ cm$^6$ s$^{-1}$) with the gas concentrations of H$_2$SO$_4$ and ELVOC. The $J_{Riccobono}$ in the default TM5-M7 parameterization is summed with the nucleation rate for binary homogenous nucleation (BHN) of H$_2$SO$_4$ and water (described by specific humidity) based on a classical nucleation theory approach




(Vehkamäki, 2002). The nucleation rate J corresponds to particles of ca. 1 nm in diameter, and the growth to 5 nm is described

using the Kerminen and Kulmala (KK) equation (Kerminen and Kulmala, 2002). The KK function in M7 determines the particle *formation rate* using available gas phase ELVOCs and $H_2SO_4$ concentrations for estimated particle survival through condensational growth, and the resulting particles then enter the explicitly modelled size distribution in the nucleation mode (Bergman et al., 2022). The $H_2SO_4$, ELVOCs, and semi-volatile organic compounds (SVOCs) can grow particles in the modal system through condensation. For the soluble accumulation mode, additional reactions of ammonia and nitric acid ($HNO_3$) can

form particulate ammonium nitrate ($NH_4NO_3$), and also methane sulfonic acid (MSA) can condense on existing ASC-mode particles (van Noije et al., 2014).

## 2.2 Detailed $H_2SO_4 - NH_3$ nucleation scheme

In accordance with Svenhag et al. (2024), this study also uses the implementation of a new approach for sulfuric acid and ammonia nucleation. Here, nucleation rates are obtained by molecular modeling and stored in lookup tables. The tables are

generated and interpolated by the J-GAIN tool (Yazgi and Olenius, 2023b). The rates are calculated by applying benchmarked, high-level quantum chemistry data to a molecular kinetics model, where the kinetic equations are solved by the state-of-the-art Aerosol Cluster and Dynamics Code (ACDC) (Olenius, 2021). The lookup table implemented in M7 gives the nucleation rate ($J$) as a function of the five variables (1) gas-phase $H_2SO_4$, (2) gas-phase $NH_3$, (3) temperature, (4) atmospheric ion production rate (IPR), and (5) cluster scavenging sink (Olenius et al., 2013). We test two $H_2SO_4 - NH_3$ schemes, created by

different quantum chemistry input data: and older data set (computed using the RICC method) and a newer dataset (computed using the DLPNO method), referred to as CLUST-High and CLUST-Low, respectively. For details, see Svenhag et al. (2024). We study the ranges produced from the *HIGH* and *LOW* inputs, for assessing the model sensitivity to ammonia. An IPR lookup table for global coverage of galactic cosmic rays and soil radon is used (Yu et al., 2019), analogous to the implementations in Svenhag et al. (2024).

## 145 2.3 EC-Earth3.4.0 simulations and emissions

We include five separate "atmosphere only" simulations for EC-Earth3.4.0-AerChem over the year 2018 with a 3-month spin-up period (i.e. from 2017-10-01 to 2017-12-31). Each simulation differs only in terms of the NPF parameterization. The five different simulation setup cases are shown in Table 1, as in Svenhag et al. (2024) with an additional "no NPF" case. All EC-Earth3 simulations have nudged meteorology from ERA-Interim for the IFS model (every 6 hours) for surface pressure, wind

divergence, and vorticity to obtain more homogenous synoptic weather between cases.

All simulations are run with emissions in accordance wth CMIP6 scenario SSP3-7.0, which includes monthly varying emissions for $SO_2$, BVOCs and $NH_3$ from 2015-2100. ELVOC and SVOC production is derived from monoterpene and isoprene reactions with OH and $O_3$ with specific rates and yields (Atkinson et al., 2006; Jokinen et al., 2015). Further descriptions for BVOCs in TM5 is stated in (Bergman et al., 2022). The isoprene and monoterpene emissions are based on the MEGAN-MACC

inventory (Sindelarova et al., 2014b). The $SO_2$ and $NH_3$ anthropogenic primary emission sources include: agriculture, energy production, industrial, transportation, residential-commercial-other, solvents production and application, waste, international





**Table 1.** The 5 EC-Earth3 simulation cases setup.

| EC-Earth3 case: | Nucleation Scheme: |
|---|---|
| Control | Equation. 1 (Riccobono et al., 2014) |
| CLUST-High | RICC2 generated Look-up table |
| CLUST-Low | DLPNO generated Look-up table |
| CLUST-Low + Riccobono | Look-up table and Equation. 1 |
| No NPF | No particle formation rates |

shipping, air, and open burning (Lamarque et al., 2010). The biogenic $NH_3$ emissions are from ocean and land with natural vegetation (Bouwman et al., 1997), and natural sources of $SO_2$ include volcanic emissions and oceanic dimethyl sulphide (DMS) that is oxidized by OH and $NO_3$. Further description of emissions for EC-Earth3-AerChem are given in van Noije

et al. (2021).

### 2.4 ADCHEM simulation and emissions

The process-based chemistry transport model ADCHEM was utilized to provide detailed modeling of gas and aerosol species and NPF, for comparisons with EC-Earth3 and field measurements obtained at the Hyytiälä and Hyltemossa research field stations during the year of 2018 (Hari and Kulmala, 2005; Neefjes et al., 2022). ADCHEM was operated as a one-dimensional

Lagrangian column model, running along air-mass back trajectories generated by the Hybrid Single Particle Lagrangian Integrated Trajectory Model (HYSPLIT) with input data on meteorology from the Global Data Assimilation System (GDAS) (Stein et al., 2015; Rolph et al., 2017). Back-trajectories were simulated seven days backwards in time with the Hyytiälä and Hyltemossa field stations as receptor points, allowing the model to source emissions of gases and particles from the urban areas, ocean and forested regions surrounding the stations. Emissions from these sources were obtained from the Copernicus

Atmosphere Monitoring Service (CAMS), including emissions from the global ocean inventory (Lennartz et al., 2017; Ziska et al., 2013; Nightingale et al., 2000; Lana et al., 2011), the global anthropogenic inventory (Granier et al., 2019) and the global biogenic inventory (Sindelarova et al., 2014a).

 New particle formation represented by an explicit coupling of molecular cluster and aerosol dynamics was obtained via the molecular cluster plugin (ClusterIn) (Olenius and Roldin, 2022). ClusterIn considered the ion-induced and neutral cluster-

ing of $H_2SO_4-NH_3$ and $H_2SO_4-DMA$ along with neutral clustering of $HIO_3-HIO_2$ and $HIO_3-DMA$. Data-sets for the $H_2SO_4-NH_3$ and $H_2SO_4-DMA$ mechanisms were computed at the DLPNO-CCSD(T)/aug-cc-pVTZ//$\omega$B97X-D/6-31++G(d,p) quantum chemical level of theory. The $HIO_3-HIO_2$ data set was calculated at the DLPNO-CCSD(T)//M06-2X method as detailed by Zhang et al. (2022). The HIO3-DMA data was calculated applying the older RICC2 method (RI-CC2//$\omega$B97X-D), and can thus be considered likely an upper estimate for the pathway. (Ning et al., 2022; Besel et al., 2020; Myllys et al., 2019).

The subsequent growth of these particles was described by considering the condensation, dissolution and evaporation of 873



organic and inorganic gas-phase species to the aerosol particle population. Particles and gasses were mixed by use of the GDAS meteorology throughout 20 horizontal layers spaced logarithmically and spanning 2100 meters into the atmosphere. The AD-CHEM model thereby attempts to reproduce the concentration of gasses and particles at different heights in accordance with the surface measurements made at both the Hyytiälä and Hyltemossa field stations. The ADCHEM simulation for Hyytiälä were only generated for May to August 2018.

## 2.5 Hyltemossa and Hyytiälä stations

The 2018 measurement data set for aerosol particle number size distributions (PNSD) at the two forest stations are produced from Differential mobility Particle Sizer (DMPS) instruments, and retrieved from the online EBAS inventory (Tørseth et al., 2012). The instruments measure aerosol diameters down to $\sim 3\,\mathrm{nm}$, but with increasing uncertainty towards the lowest diameters (Wiedensohler et al., 2012). Uncertainties in observational datasets obtained with the DMPS can also arise from factors including calibration and environmental influences, combined with diffusion and electrostatic losses for the smallest diameter aerosols. These uncertainties can potentially affect our interpretation of the PNSD at sub-100 nm. Each ramp (size distribution measurement) takes approximately 10 minutes with the DMPS. The sampled aerosol numbers are distributed into 52 (Hyytiälä) and 37 (Hyltemossa) size-bins and averaged to 1-hour arithmetic means with percentiles. Hyytiälä is the area and nearest small village where the SMEAR II station is located, and the coordinates are tabulated in Table 2.

**Table 2.** Station descriptions for the measurements used in this study (Tørseth et al., 2012).

| Station Name | Location | Instrument | Data Time Period | Lat °N | Lon °E | Altitude | Setting |
| --- | --- | --- | --- | --- | --- | --- | --- |
| SMEAR II | Finland | DMPS | 18/01/01-18/12/31 | 61.84 | 24.29 | 180 m | Forest/Rural |
| Hyltemossa | Sweden | DMPS | 18/01/01-18/12/31 | 56.10 | 13.42 | 5 m | Forest/Rural |

## 2.6 Model post-process methods

The ADCHEM and measured DMPS data sets of PNSD are presented in sectional bins. For EC-Earth3 TM5's hourly output, the modal aerosol distribution is redistributed to 100 sectional bins from 1 nm to $\sim 25\,\mathrm{\mu m}$ diameter for each 1-hour time step. Applying this modal-to-sectional conversion for every 1-hour output (same as the TM5 time step) prevents the size distribution from being unrepresentative when using longer temporal averages, which can occur when the median modal-radius output from M7 for each mode is averaged over longer time-periods. ADCHEM is run along individual air mass trajectories that arrive at the stations every 3rd hour. As EC-Earth3 and the DMPS measurements are 1-hourly values, data for occurrences when the measured DMPS is missing over more than two time steps within a single 3-hour step for ADCHEM, is removed from the analysis.



## 3 Results and discussion

The differences between the modeled and measured PNSD are presented in this section, categorized by seasonal and weekly periods for the year 2018. The seasonal median PNSD for observations and models at the two stations are shown in Fig. 1. The measured PSND at the two stations, Hyytiälä and Hyltemossa, displays a comparable seasonal variability, with lower aerosol number concentrations during winter and peak concentrations in summer. At both stations, the five EC-Earth3 cases and AD-CHEM underestimate aerosol number concentrations in summer (June to August), and overestimates in winter (December to February). When comparing the seasonal PNSD from EC-Earth3 and ADCHEM with the measurements, the ADCHEM model shows substantially better performance. EC-Earth3's median aerosol number concentrations for winter are highly overestimated at both stations (shown in Fig. 1), which is driven by primary aerosol emissions, as is clear based on the overestimation also in the no-NPF simulation. Peculiarly, the overestimation by EC-Earth during winter is much stronger at Hyytiälä than at Hyltemossa. In springtime at both stations, the EC-Earth3 simulations have the highest disparity between model runs with the different NPF schemes. At such boreal conditions, springtime is favorable to nucleation as the period is photochemically active, but temperatures are not too high. The default $ELVOC-H_2SO_4$ (Riccobono) nucleation depend on biogenic emission from vegetation, and the NPF is significantly lower in the control case during spring before the initiated growing season. On the other hand, agricultural $NH_3$ emissions peak in spring due to application of fertilizers. The inclusion of $NH_3-H_2SO_4$ nucleation in spring, as modeled by the CLUST scheme cases, improves the agreement between the model and observations during this period. Shown in the hourly distributions and the seasonal median (Fig. 1 and Fig. 3), the model's ability to capture NPF-days is improved, and the results show more accurate magnitude of the nucleation events and the subsequent Aitken-mode number concentration.

The total aerosol concentrations during summer (Jun-Aug) is underestimated in EC-Earth3 shown for all model cases in Fig. 1, and similarly in an hour-to-hour comparisons during August and July shown in Fig.5. This insufficiency in NPF and growth could be attributed to low availability of condensable vapors, or a limitation for the model growth mechanisms in EC-Earth3, since ADCHEM can produce higher levels with similar nucleation schemes. The highest levels of available ELVOCs occurs in summer, and for these months the default $ELVOC-H_2SO_4$ will produce higher surface particle formation rates at the model surface level compared to the CLUST-High and CLUST-Low case (Fig. 6). However, the CLUST cases still have greater Aitken-mode aerosol number concentrations at the surface. This can be explained by CLUST's substantially greater aerosol formation rates in the overlying grids (or possibly in neighboring grids). These additional aerosols can then descend (or move laterally) to the surface grid representing the station.

The effect of the CLUST scheme implementation on the cloud radiative effect (CRE) and the direct aerosol radiative effect (DRE) is shown in Table A1 for the different seasons at the two stations. Again, the springtime period exhibits the largest model change from implementing CLUST. However, as local conditions can change by advected clouds or aerosols formed elsewhere, we do not discuss the radiative changes further. The multi-year global CRE and DRE effects of using CLUST in EC-Earth3 are elaborated on in Svenhag et al. (2024).





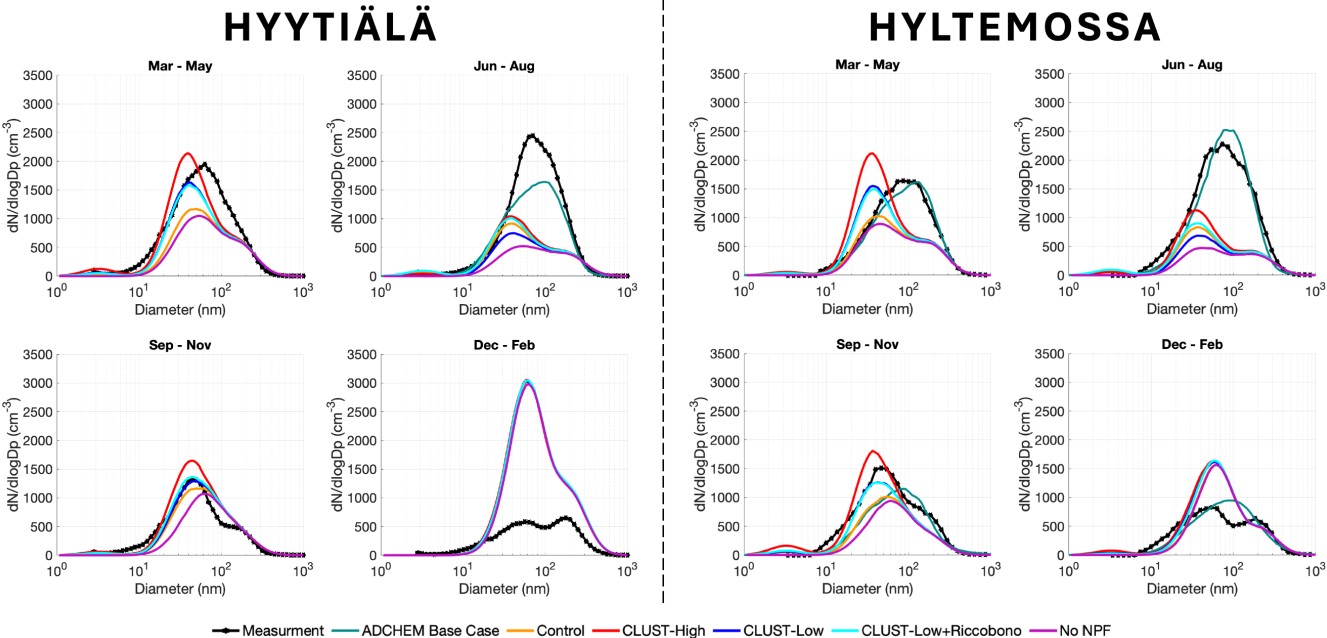

**Figure 1.** The seasonal median size distribution for the five EC-Earth3 variants, the ADCHEM simulations, and the observations. Note, the ADCHEM model only generated output from May to August for Hyytiälä and thus is only shown for Jun-Aug.

## 3.1 Vertical distribution

For the 5 EC-Earth3 model variant cases, there is a great difference in the vertical particle formation rate profiles from the second model layer upward (approx 100 m altitude and above) when comparing $ELVOC-H_2SO_4$ and $NH_3-H_2SO_4$ simulated rates, as shown in Fig. 2d. This occurs up to $\sim$400 hPa, then the water-$H_2SO_4$ (BHN) nucleation prevails in the upper troposphere similarly for all cases. The cause of this model difference is related with the available precursor gases at different altitudes, as shown in Fig. 2a-c. The decline in particle formation rates in the control case is explained by the available modelled ELVOC gas concentration rapidly declining with altitude, and consequently the $ELVOC-H_2SO_4$ nucleation is limited to only near-surface NPF in the boundary layer. The particle formation rate for the four EC-Earth3 cases is represented over the seasons (as a daily mean) at Hyytiälä is shown in Fig. A1. The CLUST schemes are producing the highest formation rates in spring and autumn at high altitudes. There are occasions when the formation rates for the Control case are occurring at altitudes between 800–400 hPa, but these rates are almost entirely produced by the model from only BHN of water$-H_2SO_4$. The vertical mean distribution was similarly shown for the global average in Svenhag et al. (2024), and the resulting EC-Earth3 distributions in this study gives resembling case differences for the annual mean vertical profile between CLUST cases and the control case at the Hyytiälä and Hyltemossa stations. The evaluated weekly cases for each season also highlight the variations in EC-Earth3 between model level 1 (surface) and level 2 ($\sim$100 m) for aerosol formation rates, as well as ELVOC, $H_2SO_4$, and $NH_3$ concentrations (as presented in the following section).





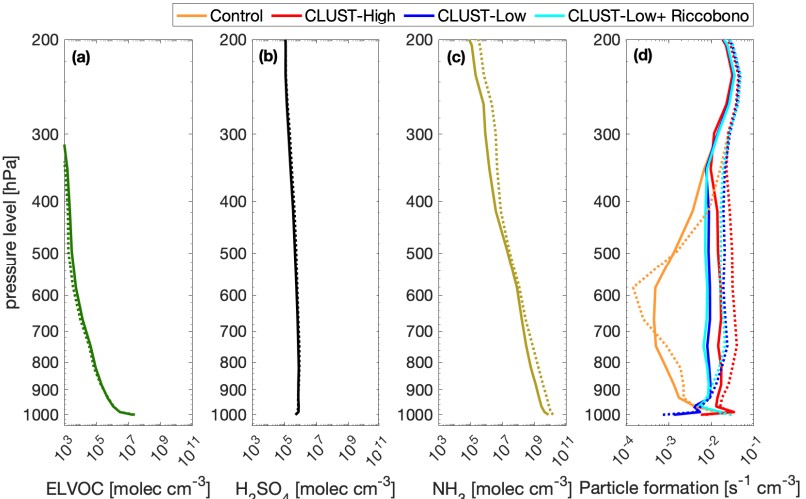

**Figure 2.** EC-Earth3's modelled annual mean vertical precursor (a) ELVOC, (b) $H_2SO_4$ (c) $NH_3$ concentrations, and (d) the particle formation rate for 2018. The bold line represent Hyytiälä and dotted is Hyltemossa station. The gas concentrations in a-c are from the No-NPF case, and the median differences between EC-Earth3 model cases would not be visible in this figure.

### 3.2 Seasonal new particle formation

In this section, we evaluate hourly model output over separate weeks with observed NPF events from March to October, with most focus on the springtime. Firstly, two weekly spring cases in March and April are shown for the modelled aerosol size distributions and observed springtime new-particle formation events measured by the DMPS at the two stations (Fig. 3). As expected, the more detailed chemistry and sectional aerosol scheme in ADCHEM captures and reproduces these example events with better resemblance to observations compared to the EC-Earth3 model cases. In Fig. 3 (left panel) both observed

or modelled NPF at Hyytiälä is initially (March 12[th]-13[th]) likely suppressed by a high concentration of background Aitken and accumulation mode aerosols. EC-Earth3's mechanisms halts NPF on these days from the sinks produced by the larger mode aerosols, since the concentration of precursor gases are at similar levels to the following days when NPF is occurring, shown in Fig. 4. Afterward, four exemplary NPF and growth (banana) events at Hyytiälä are measured the succeeding days of March 14[th] to 17[th]. These four NPF events modelled by EC-Earth3 cannot be physically replicated (with a banana shape)

due to the limitations of how aerosols grow and move between the modes. Nonetheless, the resulting total concentration in the nucleation and Aitken mode agrees better with observations when using the CLUST scheme compared to using only the default ELVOC$-H_2SO_4$ nucleation. Comparing the modelled "No NPF" case with the Control case for Hyytiälä in Fig. 3 (left panel) shows the weak visible nucleation in the control case having little to no effect on the Aitken mode number concentrations.

Fig. 3 (right panel) shows the modelled Hyltemossa station during seven days in April, and it presents three measured
typical nucleation (banana) events which are also captured differently by the model cases on April 6[th]-8[th]. The ADCHEM model captures these three events fairly well, but still show some deficiencies. From EC-Earth3, the resulting events are more



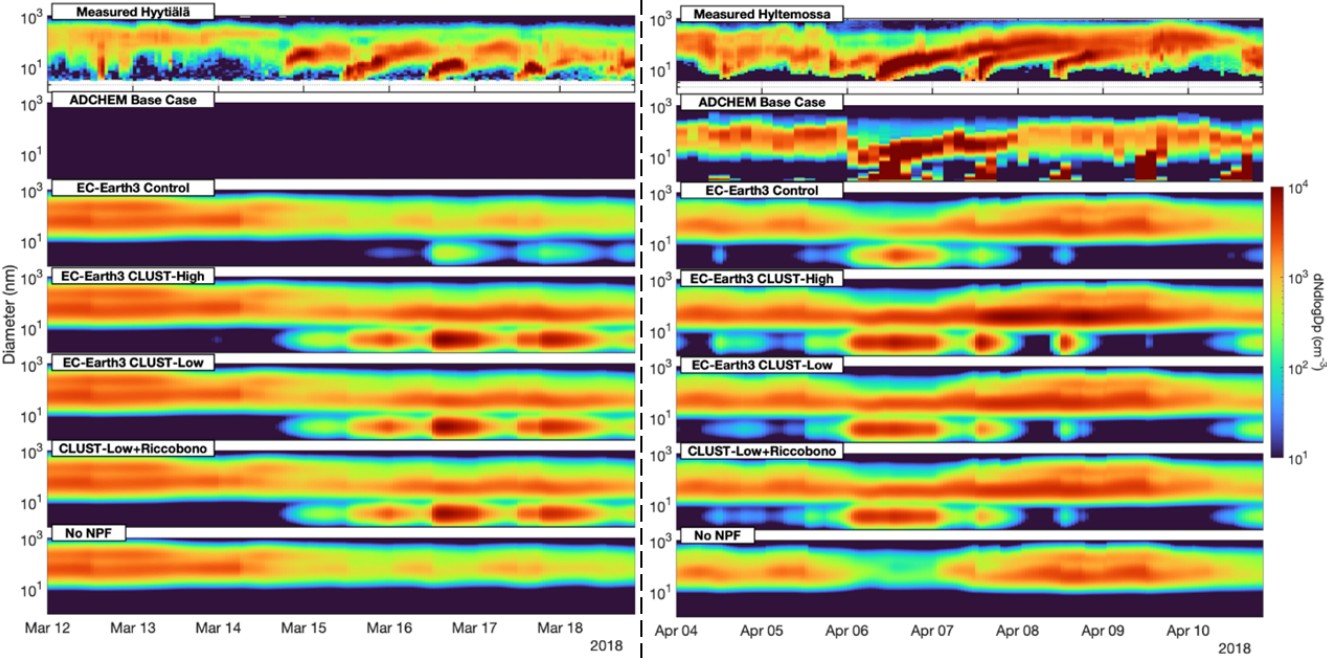

**Figure 3.** Surface aerosol number size distribution over the springtime for the 5 modelled EC-Earth3 cases, with the DMPS measured aerosols at Hyytiälä and Hyltemossa. ADCHEM simulations have no available hourly data outside the Summer months for Hyytiälä.

irregular. The first event on April $6^{th}$ is captured, but underestimated in intensity and growth, with a continuously high NUS concentration throughout the day. The EC-Earth3 CLUST cases gives aerosol number concentrations closer to the measured concentrations for the events on the $6^{th}$-$8^{th}$ compared to the default control case. This strong event on April $6^{th}$ shown in Fig.

4 i, j result in large differences for $H_2SO_4$ gas-phase concentrations between EC-Earth3 cases. Significantly more $H_2SO_4$ is consumed on this day in the CLUST scheme cases. Additionally, ELVOC concentrations are low during this time (Fig. 4 k), while $NH_3$ concentrations are higher on the $6^{th}$ compared to the average spring levels shown in Fig. A5. The variation in NPF event strength for this April case in EC-Earth3 at Hyltemossa closely resembles the weekly autumn case observed in October at the same station (shown in Fig. A6 and Fig. A7).

In this section, we include model results from the $2^{nd}$ model height-level in TM5 to highlight the significant changes in formation rates and gas concentrations between model layers 1 and 2, with the differences for the nucleation schemes. Fig. 4 demonstrates the differences in the scheme's nucleation rates and gas concentrations for the same spring week as in Fig. 3. For the $H_2SO_4$ during this week, concentrations are generally higher in the second model level compared to the surface level. The days when the different nucleation scheme cases are diverging in $H_2SO_4$ concentrations from the "No NPF" case

shows how the different cases are consuming $H_2SO_4$ to form and grow new particles, e.g. shown in Fig. 4 i, j on April $6^{th}$. The





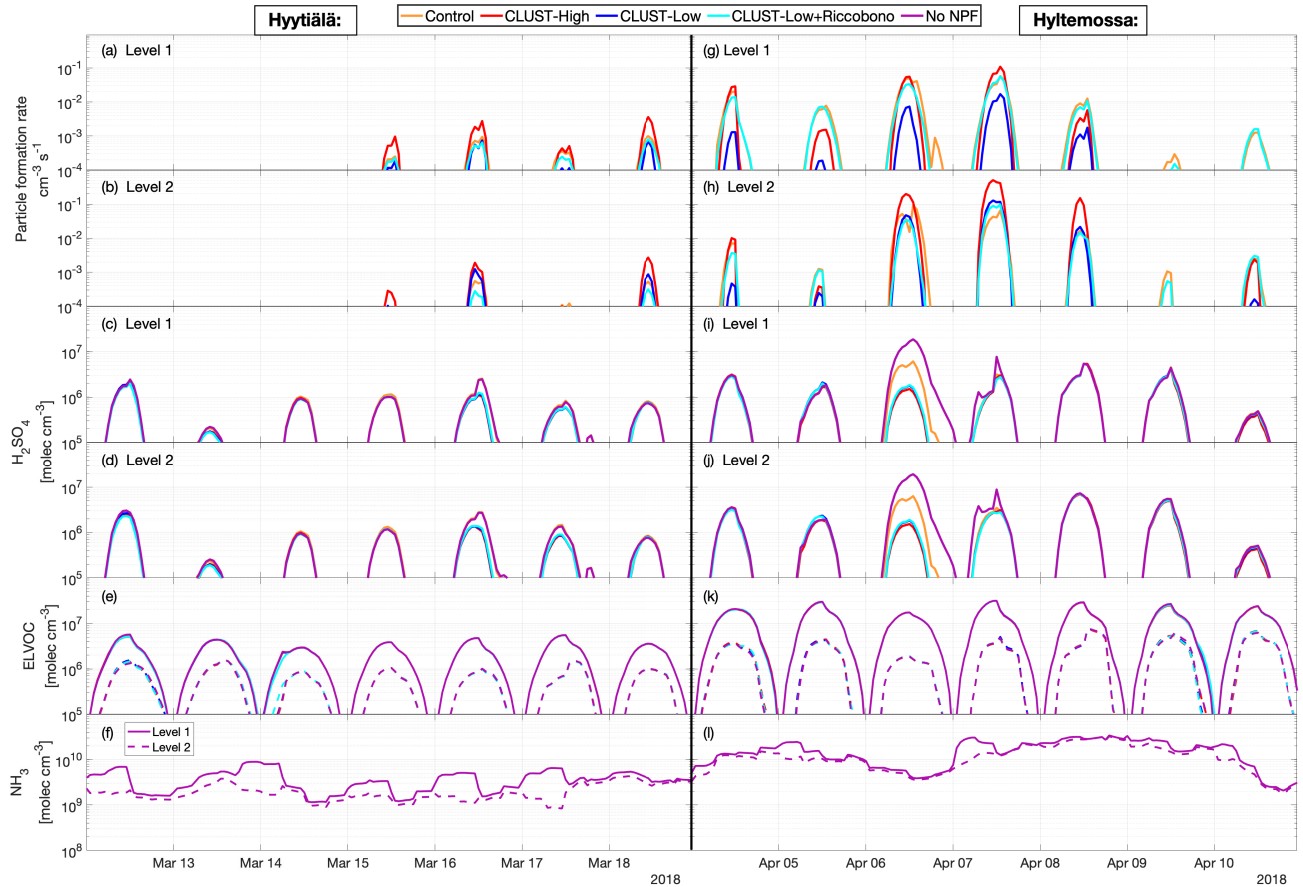

**Figure 4.** One week of modelled layer 1 and layer 2 Hyytiälä and Hyltemossa springtime cases showing (a, b, g, h) daily maximum particle formation rate, (c, d, i, j) $H_2SO_4$, (e, k) ELVOC, (f, l) and $NH_3$ gas concentration. The ELVOC and $NH_3$ concentrations are from the No-NPF case. The missing particle formation rates below $10^{-4}$ in panel a, b, g, and h are considered practically zero.

ELVOC concentrations decrease by a factor of approximately 10 between levels 1 and 2 (Fig. 4 e, k), a pattern also observed for the SVOC concentrations (not shown). The springtime modelled ELVOC abundance at Hyltemossa compared to (more northern) Hyytiälä location drops by a factor of 10 at Hyytiäla, while contrarily, the measured total aerosol concentrations are slightly higher at Hyytiäla. This further suggest that we are lacking some model emissions or chemistry in EC-Earth3 around the station. Yet, underestimated summer PNSD at Hyytiälä is shown for both ADCHEM and EC-Earth3, which indicates some parameterizations and/or emissions for SOA might be missing at Hyytiälä in both models.

We evaluate an additional two weeks for the summer period at Hyltemossa and Hyytiälä (Fig. 5). It shows large underestimations in EC-Earth3 (all cases) for NPF magnitude and growth. The control case with ELVOC-$H_2SO_4$ results in higher particle formation rates (compared to CLUST) at the surface during this period at both locations, as shown in Fig. 6 a, g. Despite the higher summertime surface formation rates for ELVOC-$H_2SO_4$ nucleation, this case still yields a lower median surface aerosol





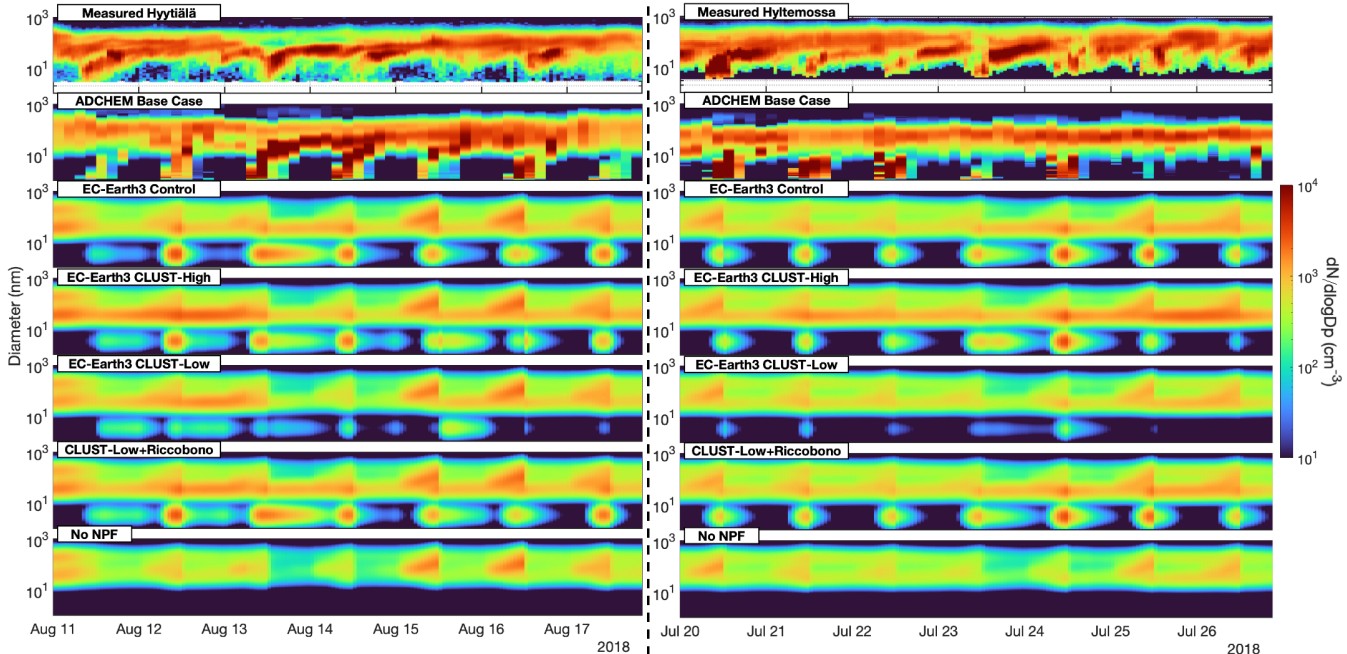

**Figure 5.** Surface aerosol number size distribution over the summer week for the 5 modelled EC-Eart3 cases, with ADCHEM and the DMPS measured aerosols at Hyytiälä and Hyltemossa.

number concentration for the summer season compared to the CLUST-High case, shown in Fig. 1. The particles produced in overlying and neighboring grids for CLUST-High is likely responsible for the resulting higher median summer surface concentrations here. Similar to the hourly results in Fig. 4, the annual median particle formation rates in Fig. 2 show significantly higher particle production in the second model layer compared to the surface layer.

## 3.3 Further discussion

Expanding on the vertical particle formation rates, the detailed ADCHEM model results suggest that the particle formation rate between the surface and 2000 m (800 hPa), shown in Fig. A2, exceeds what EC-Earth3's $ELVOC-H_2SO_4$ and $water-H_2SO_4$ nucleation (control case) produces ($< 10^{-3}\,s^{-1}cm^{-3}$). These results further support the inclusion of $NH_3-H_2SO_4$ NPF (like the CLUST table) as a nucleation pathway for the lower troposphere, especially above 100 m. However, while the newly-implemented ion-dependent $NH_3-H_2SO_4$ mechanism is an important NPF pathway, also other mechanisms and species may contribute. Studies have suggested the potential importance of amine and iodine enhanced nucleation (Wollesen de Jonge et al., 2024; Zhao et al., 2024). However, the sources and atmospheric concentrations of such species involve uncertainties, which propagate to the predictions of their effects on NPF. Further, our current CLUST scheme does not include the effects of e.g. hydration or nitric acid ($HNO_3$) on $NH_3-H_2SO_4$ driven NPF, due to the lack of complete molecular data sets for calculating the formation rates, and testing these effects could be beneficial (Zhao et al., 2024). Additionally, validating TM5's



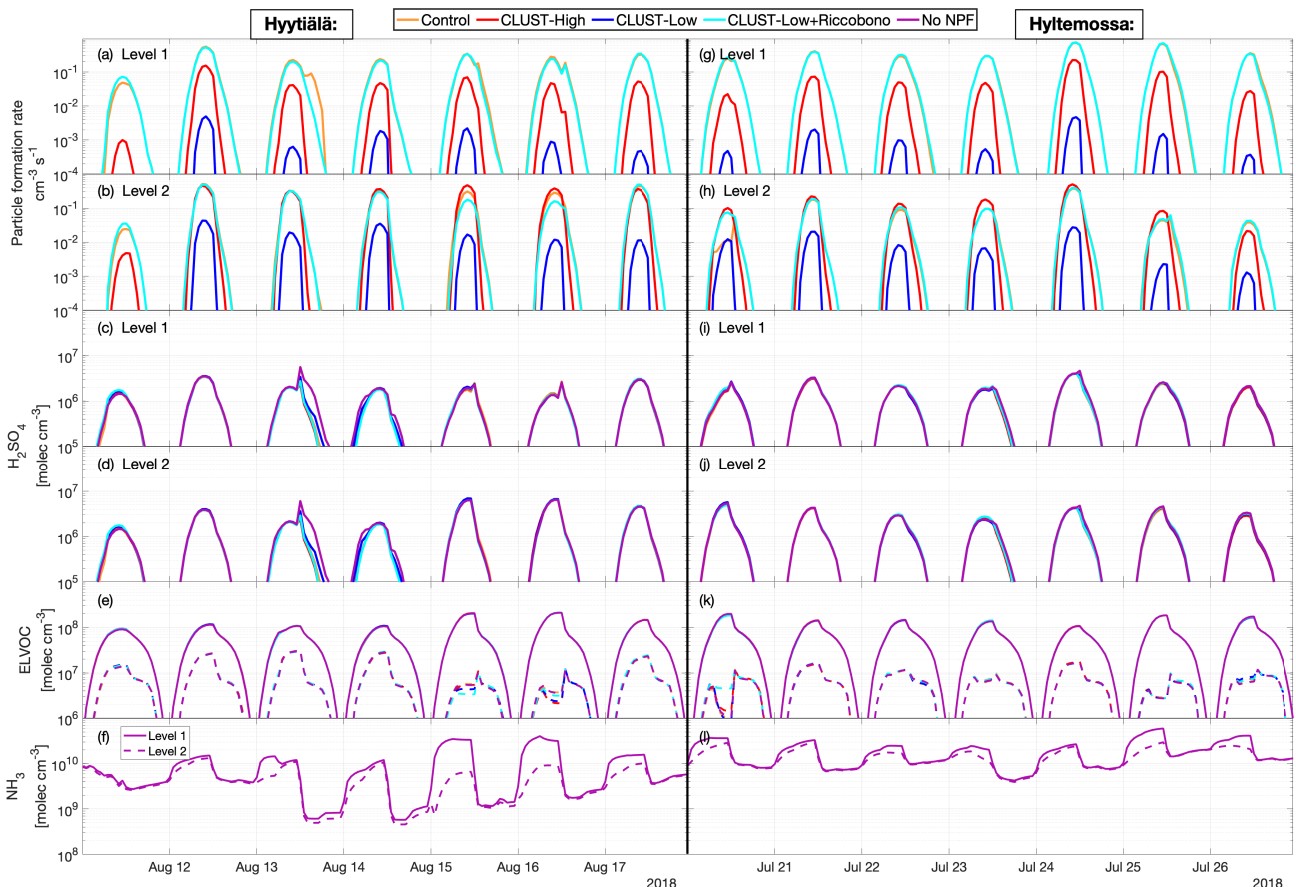

**Figure 6.** One week of modelled layer 1 and layer 2 Hyytiälä and Hyltemossa summer periods showing (a, b, g, h) daily maximum particle formation rate, (c, d, i, j) $H_2SO_4$, (e, k) ELVOC, and (f, l) $NH_3$ gas concentration. The ELVOC's y-axis scale magnitude is $\times 10$ greater in reference to Fig. 4i,j.

very low vertical distribution for the BVOCs concentrations above the surface model layer, could improve the boreal (and global) representation of aerosol in EC-Earth3. Further improvements could be achieved by incorporating detailed model comparisons to refine the fixed ELVOC and SVOC yields beyond the two categorical species currently used in TM5. Such enhancements could better capture VOC abundance and thereby improve predictions of particle formation rates and growth. Assuming the more sophisticated chemistry from ADCHEM would produce "significant enough results" with respect to the increased computational burden.

It should also be noted that the scheme for $\mathrm{organic} - H_2SO_4$ nucleation is simple, with no dependencies on temperature, scavenging sink or IPR (Eq. 1). Because of this, the modeled nucleation rate for this pathway does not fully correspond and/or respond to the ambient conditions and their changes.



Comparing the modelled mass concentrations in ADCHEM versus the EC-Earth3 versions, revealed that the surface organic aerosol mass is significantly higher for ADCHEM, especially during the month of July, shown in Fig. A3. This disparity in organic aerosol mass could be due to a combination of the factors mentioned above, e.g. the models' emission inventories, parameterization for SOA formation, and/or chemistry of the organic precursors. Noting the ADCHEM trajectories shows the prevailing air-masses are coming from continental areas with high organic emissions during this period (Wollesen de

Jonge et al., 2024). Evaluating and refining the available condensable organic (and inorganic) vapor for secondary aerosol formation and growth in EC-Earth3 is crucial for future studies, as the current nucleation and growth schemes lead to significant underestimations of summer Aitken and accumulation mode concentrations at boreal stations. The overestimation by EC-Earth3 in winter might be due to the coarse resolution of EC-Earth3, which entails that anthropogenic emission which are quite far from the station are still emitted within the same grid box.

A significant model uncertainty produced by TM5 is revealed in the hourly representations of near-surface aerosol concentrations. This is visible in Fig. 5 at noon (12:00), and occurs due to TM5's 6-hourly meteorological dynamics retrievals from IFS. The IFS dynamics controls the boundary layer height and consequential vertical mixing, and will maintain the 6:00 conditions up until new values are retrieved at 12:00 which causes an abrupt change in surface gas and aerosol concentrations following this time step. At the two stations, this effect is most prominent at noon in summer, when the boundary layer experiences its

maximum fluctuation during the day. Similarly, at midnight (24:00) the surface aerosol concentration starts accumulating much faster as the boundary layer height is drastically decreased when TM5 retrieves the new IFS variables (from previous 18:00 conditions). The $NH_3(g)$ concentrations (and likely many gas compounds) are also dramatically influenced by these shifts at noon and midnight, shown in Fig. 6. The unrealistic dynamics produced by TM5, resulting from this model assumption, introduce greater uncertainty in direct comparisons with hour-to-hour NPF events and aerosol populations in this study. It would

also add uncertainty to how aerosols near the surface potentially influences cloud-aerosol interactions in EC-Earth3 within this 6-hour window.

## 4    Conclusions

The new CLUST implementation of $H_2SO_4-NH_3$ nucleation accounts for more spring and autumn NPF in the boreal regions, where the default EC-Earth3 case is underestimating aerosol numbers. Comparing the CLUST cases versus the control run;

the total 2018 aerosol number concentration at Hyytiälä increased by 53 % for CLUST-High, 12 % for CLUST-Low, and 22 % for CLUST-Low+Riccobono. At Hyltemossa it increased by 91 % for CLUST-High, 28 % for CLUST-Low, and 34 % for CLUST-Low+Riccobono. From the resulting aerosol PNSD during spring, summer and autumn, we suggest the use of CLUST-Low+Riccobono version with $H_2SO_4-NH_3$ and $H_2SO_4-organic$ pathways as a default option in future EC-Earth3 studies to account (in principle) for both nucleation mechanisms in EC-Earth3. ADCHEM uses the same chemistry

data for the $H_2SO_4-NH_3$ pathway as implemented in the CLUST-Low scheme. The CLUST-Low scheme applies the most recent quantum chemistry methods for the $H_2SO_4-NH_3$ nucleation rate and is considered to have more realistic sensitivity to ammonia compared to the previous scheme (CLUST-High).



Furthermore, the ADCHEM model results and the global NPF modelled by Zhao et al. (2024), suggests the mean atmospheric new particle formation rate is higher than $10^{-2}$ s$^{-1}$cm$^{-3}$ between the second model layer in TM5 ($\sim$100 m) and the

upper troposphere. Our new CLUST nucleation mechanism accounts for more NPF in this vertical region, especially in the northern mid-latitudes, which agrees with the detailed modelling in ADCHEM and Zhao et al. (2024). The increase in Aitken mode particles may also be attributed to more nucleation occurring at higher altitudes. This further suggests the inclusion of the $H_2SO_4-NH_3$ NPF pathway in Earth System Modelling to cover nucleation and production of particles in low ELVOC environments and at higher altitudes, which have significant influence on model clouds properties and radiative effects (Svenhag

et al., 2024).

This study further highlights the potential false representation from using solely annual medians when evaluating EC-Earth3's (and other ESMs) performance due to significant seasonal variations. An annual average of the two modeled stations in this study would merge the model's significant underestimations of summer aerosol concentrations with its substantial overestimation in winter, resulting in a balanced median value. As the climate forcing from aerosols have large seasonal variations,

it would be crucial to have a representative seasonal model depiction of aerosol formation and growth. The underestimated aerosol concentrations for EC-Earth3 in the summer could have a significant impact on the radiative outcome with regard to the high sensitivity between CRE and sub-100 nm aerosols in EC-Earth3. The observed mechanisms in these regional environments are not necessarily transferable to different environments, which makes comparisons towards modelled NPF effects on a global scale challenging. Another challenge is introduced by the uncertainty from the TM5's 6-hourly exchange of meteo-

rological dynamics from the IFS model, which alters the aerosol and precursor gas concentrations and conditions significantly near the surface at noon and midnight.

We have demonstrated that the detailed evaluation of seasonal boreal new-particle formation in the EC-Earth ESM can be accomplished by both representative field observations, and detailed aerosol simulations (ADCHEM). This combination aims to recognize the factors affecting the ability of the ESM to capture NPF trends. Such analysis can be applied (1) to

assess seasonal trends and possible reasons for season-wise biases, and (2) to separate different error sources stemming from fundamental model schemes, e.g. inclusion/exclusion of given NPF mechanisms, and from ESM model approximations for e.g. aerosol size distribution.





**Appendix A**

**Table A1.** The IFS model 2018 seasonal mean of the cloud radiative effect (CRE) and aerosol direct effect (DRE) over the stations. The first two columns show the control case, while the other columns shows the CLUST cases subtracted with the Control case. Bold numbers are given for significance with Student's t-test (95%)

| | Control run | | CLUST-High - Control | | CLUST-Low - Control | | CLUST-Low + Ricco - Ctrl | |
|---|---|---|---|---|---|---|---|---|
| | CRE | DRE | CRE | DRE | CRE | DRE | CRE | DRE |
| Hyytiälä Spring | -13.89 | -2,33 | **-2.18** | -0.0376 | **-1.27** | -0.0182 | -1.68 | -0.0090 |
| Hyltemossa Spring | **-30.15** | **-5,67** | **-3.51** | **-0.1367** | -1.28 | **-0.0355** | -0.77 | -0.0391 |
| Hyytiälä Summer | **-42.21** | **-5.97** | +0.00 | -0.0305 | +1.23 | -0.0086 | +0.58 | **-0.0190** |
| Hyltemossa Summer | **-27.74** | **-5,39** | -2.92 | -0.0271 | -0.14 | +0.0102 | -0.65 | -0.0120 |
| Hyytiälä Autumn | -4.06 | **-3.12** | -0.53 | +0.0146 | +0.23 | **+0.0212** | +0.41 | +0.0106 |
| Hyltemossa Autumn | -4,86 | **-3,37** | -0.92 | **-0.0265** | **-1.02** | -0.0246 | -0.93 | **-0.0295** |
| Hyytiälä Winter | +10.58 | **-0,55** | +0.04 | -0.0048 | -0.45 | -0.0033 | +0.09 | +0.0025 |
| Hyltemossa Winter | +8,92 | **-2,01** | -0.29 | **-0.0261** | **-0.52** | **-0.0181** | **-0.77** | -0.0340 |



**Figure A1.** Vertical distribution of daily mean particle formation rates in the four EC-Earth3 cases at Hyytiälä for 2018, March to October.



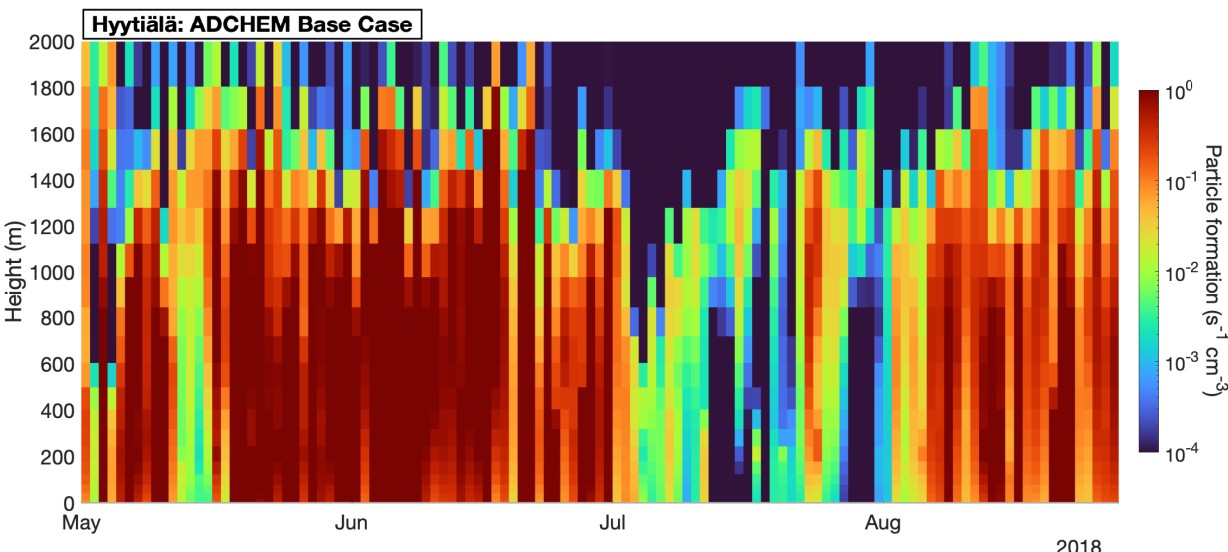

**Figure A2.** Vertical distribution of daily mean particle formation rates in ADCHEM Base Case at Hyytiälä 2018, May to August



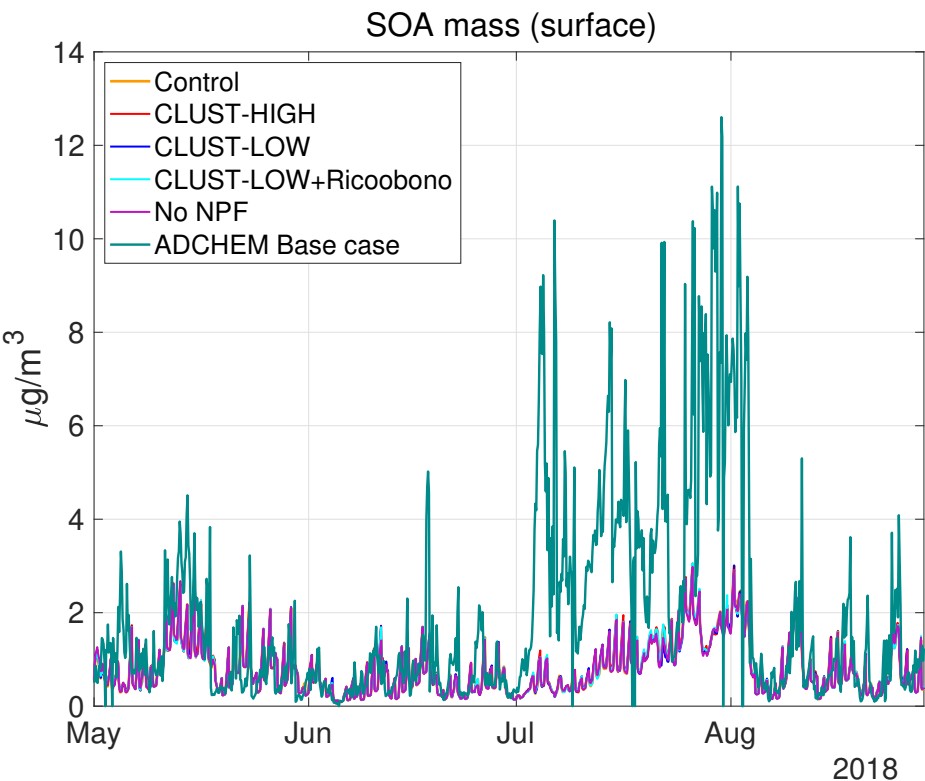

**Figure A3.** Modelled mass concentration of particulate organic matter (SOA) for EC-Earth3 cases and ADCHEM at Hyytiälä 2018, May to August



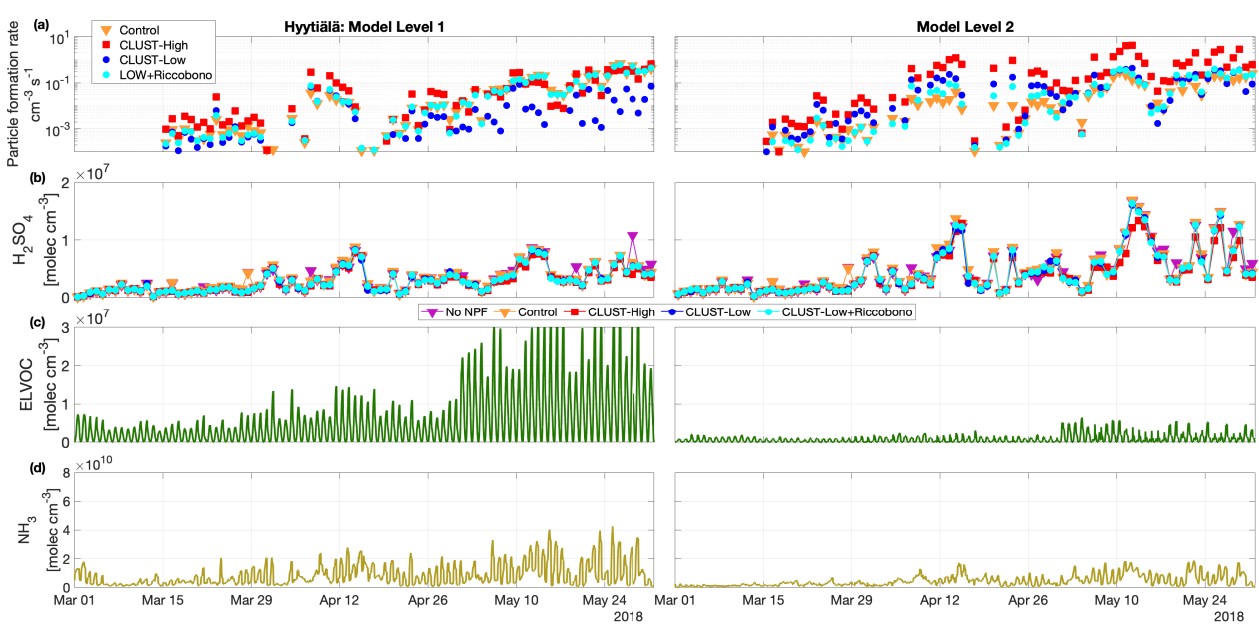

**Figure A4.** EC-Earth3 modelled layer 1 and layer 2 Hyytiälä springtime (a) daily maximum particle formation rate (b) $H_2SO_4$ (c) the No-NPF case ELVOC and (d) the No-NPF case $NH_3$ gas concentration.



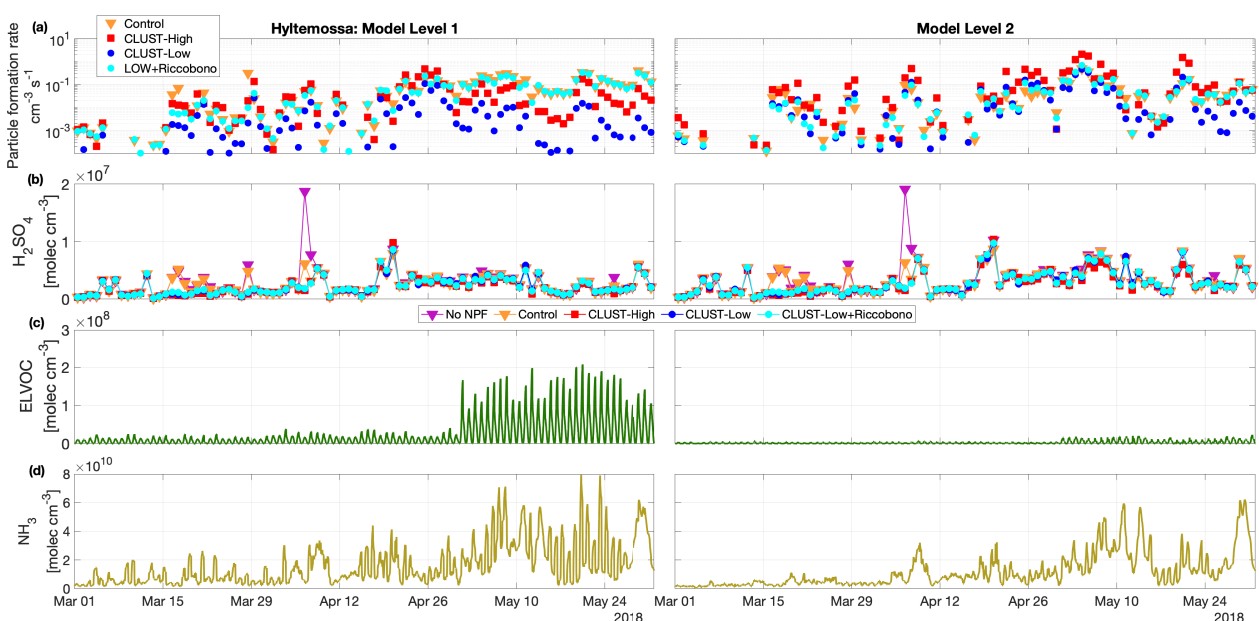

**Figure A5.** Same as Fig. A4 but for Hyltemossa. The ELVOC's y-axis scale magnitude in (c) is ×10 greater here in reference to Fig. A4c.




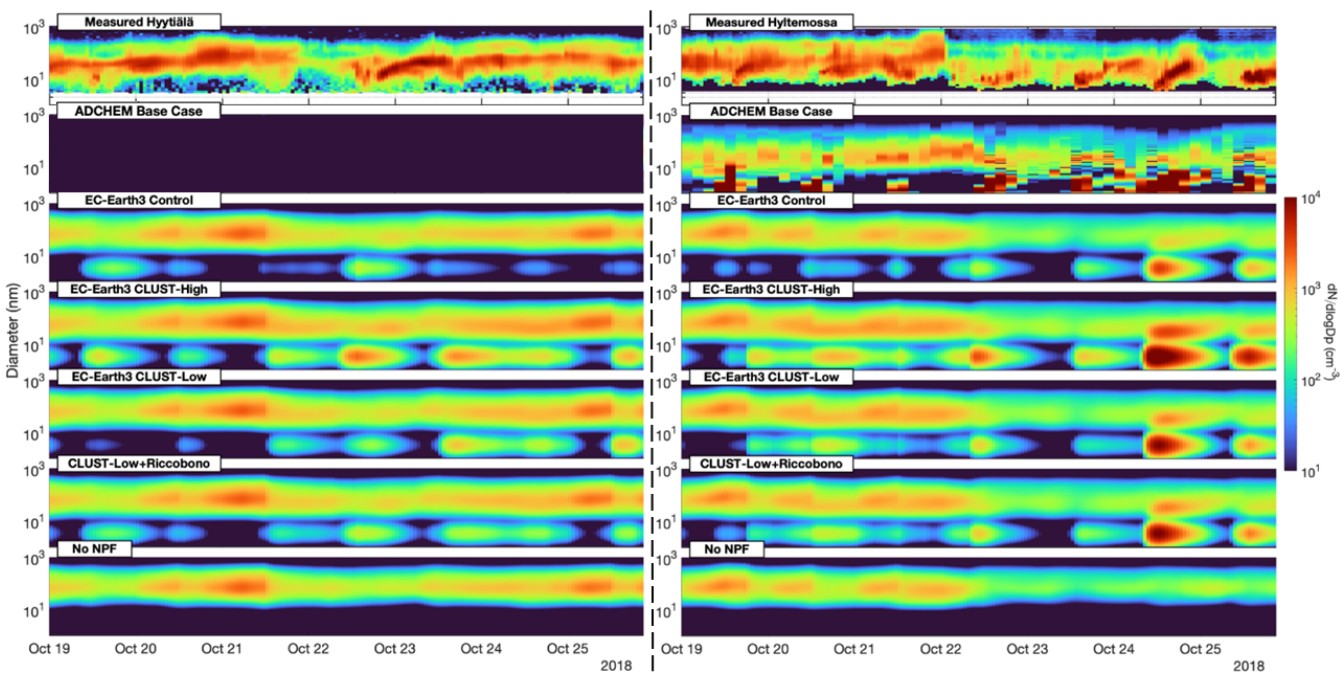

**Figure A6.** Surface aerosol number size distribution over the Autumn for the 5 modelled EC-Eart3 cases, with ADCHEM and the DMPS measured aerosols at Hyytiälä and Hyltemossa.



**Figure A7.** One week of modelled layer 1 and layer 2 Hyltemossa autumn period showing (a, b, g, h) particle formation rate, (c, d, i, j) $H_2SO_4$, (e, k) ELVOC, and (f, l) and $NH_3$ gas concentration. The ELVOC's y-axis scale magnitude is $\times 10$ greater in reference to Fig. 4i,j.

*Code and data availability.* Model code for TM5-MP version 1.2 with implemented CLUST look-up table is found at Svenhag (2024a).
The model output datasets are found in Svenhag (2024b), post-process scripts are located in Svenhag (2024c). Codes for the J-GAIN v1.0 generator and the interpolator used for the CLUST look-up table in the experiments can be found at Yazgi and Olenius (2023a). Resources for the IPR lookup table can be found in Yu (2019). The DMPS and SMPS measurement products at the stations are available for downloaded at https://ebas-data.nilu.no/Default.aspx.

*Author contributions.* CS, MS, and PR designed the research idea, CS performed the EC-Earth3 simulations and post process handling.
RWD and PR provided ADCHEM base case data and its model description. TO and DY developed the CLUST model, and CS performed the EC-Earth3 implementations of CLUST. SMB contributed with discussions for the analysis.



*Competing interests.* The authors declare that they have no conflict of interest.

*Acknowledgements.* The computations and data handling were enabled by resources provided by the National Academic Infrastructure for Supercomputing in Sweden (NAISS) and the Swedish National Infrastructure for Computing (SNIC) at Tetralith, partially funded by the Swedish Research Council through grant agreement nos. 2022-06725 and 2018-05973. The authors thank the Centre for Scientific and Technical Computing at Lund University (LUNARC), partially funded by the Swedish Research Council through grant agreement nos. 2022-06725 and 2018-05973.

The authors thank Markku Kulmala and Pasi Aalto for DMPS measurements obtained at SMEARII, and Adam Kristensson for DMPS measurements obtained at Hyltemossa station. Thanks to the work of the technical staff and scientists in guaranteeing the quality of the data is recognized from the ACTRIS program. This work is supported by the Strategic Research Area "ModElling the Regional and Global Earth system", MERGE, funded by the Swedish government.

*Financial support.* This project was funded by the Swedish Research Council for Sustainable Development, Formas (project no. 2018-01745-COBACCA, grant no. 2018-01745). This research has also been supported by Formas grant 2019-01433, the Swedish Research Council (Vetenskapsrådet, grant nos. 2019-05006, 2019-04853, and 2022-02836), the EU Horizon Europe project AVENGERS (grant no. 101081322), the Crafoord Foundation (grant no. 20210969), the EU Horizon project PAREMPI (grant no. 101096133), the European Union's Horizon 2020 research and innovation program (project FORCeS under grant agreement no. 821205), the European Research Council (consolidator grant INTERGRATE no. 865799), and the Knut and Alice Wallenberg Foundation (Wallenberg Academy Fellowship project AtmoRemove, grant no. 2015.0162).

The publication of this article was funded by the Swedish Research Council, Forte, Formas, and Vinnova.



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
