# Peer review of "Seasonal differences in observed versus modeled new particle formation at two European boreal stations"

_EGUsphere, 2024_

## Author Comment (AC2)

**Author Response to Christina Willamson**

We thank the reviewers for their thorough and constructive comments, which have helped to improve the quality and clarity of our manuscript. We have carefully considered all suggestions and made corresponding revisions in the manuscript. Below, we provide detailed responses to each comment, with changes clearly indicated in the revised version of the manuscript.

Reviewer comments are reproduced in **bold**, and our responses follow each comment in **plain text**. Line numbers refer to the revised manuscript unless otherwise stated.

- Start of CJW Comments: -

**My review here below is formatted as direct answers to the questions posed for consideration in ACPs review criteria.**

- **Does the paper address relevant scientific questions within the scope of ACP?**

**This paper addresses the questions of how to represent aerosol nucleation and growth in global models, and how to evaluate these representations, both of which are within the scope of ACP.**

- **Does the paper present novel concepts, ideas, tools, or data?**

**The paper presents novel model outputs to compare with field observations, as well as a novel evaluation of EC-Earth3 using both observations and a more complex chemical model. In particular, this study's use of both observations and detailed chemistry process model to understand why EC-Earth and observations differ, and to check things are "right for the right reasons" is very valuable.**

- **Are substantial conclusions reached?**

**Substantial conclusions are reached regarding the need to include ammonia nucleation mechanisms in EC-Earth3 to better represent nucleation and growth in European boreal forest, and the necessity of evaluating seasonal and even hourly outputs to properly assess model representations of aerosol nucleation and growth.**

**Are the scientific methods and assumptions valid and clearly outlined?**

**While much of the method used and most of the assumptions made are valid and clearly outlined, some of the assumptions do not seem valid or well justified. I will detail the areas I consider problematic in this respect here below:**

**The introduction refers to Hyytiälä and Hyltemossa stations as "not heavily impacted by anthropogenic influence" (line 76-81) but then goes on to discuss higher levels of NH3 and H2SO4 at Hyltemossa due to anthropogenic activity. This suggests that at least the Swedish station is influence by anthropogenic activity. We are also aware of anthropogenic activity influencing NPF at the Finnish station, through the presence of NH3 and elevated H2SO4. Since NPF mechanisms and growth of small particles are sensitive to small concentrations of NH3 for example, we cannot justify these as representative natural sites. This does not**

**invalidate the methodology, but greater care should be taken to present the conditions of these measurement sites and not try to present them as representative of the broader northern midlatitude region. Also, this study specifically addresses NH3 NPF, so if the sites were not anthropogenically influence this would be a problem since we would not expect any influence of NH3 in that case.**

We agree that both Hyytiälä and Hyltemossa experience varying degrees of anthropogenic influence, and it is not accurate to characterize them as purely "natural" or free from such effects. We have therefore revised the wording in the Introduction to more accurately reflect the conditions at these sites. The updated sentence now reads: "The surrounding area is primarily spruce forest, with some anthropogenic and agricultural influences, particularly near the southern station.". We also now clarify in the text that the sites are not intended to serve as broad representative of all northern midlatitude environments, but are instead a depiction of forested regions with both biogenic and anthropogenic inputs.

**This problem of representativeness appears again in the conlusions ( line 355). Text states that CLUST mechanism accounts for more NPF between 100m altitude and upper troposphere "especially to northern mid-latitudes". Firstly, since only data and modelling were presented for northern mid-latitudes, it is not justified to imply these results are applicable elsewhere. Secondly, it is not justified that the 2 presented site are typical for northern mid-latitudes. Indeed, when we consider that the major conclusion is the need to include ammonia nucleation and growth at the 2 sites presented, we see that we are dealing with anthropogenic influence that would likely not apply to the same extent when considering boreal forest sites in north America. The EVOC representation from Riccobono may be more applicable to the emissions from European boreal forest than north American or Siberian boreal forest, and then of course there are marine regions in northern mid latitudes as well.**

That is a good point regarding the representativeness of our study sites and the scope of our conclusions. We agree that it is not appropriate to imply broader applicability beyond the specific conditions of the two sites presented, particularly given the known variability in emissions and anthropogenic influence across different northern mid-latitude regions. In response, we have revised the text in the Conclusions (and the title) to more accurately reflect the scope of our findings. The sentence now reads: "Our new CLUST nucleation mechanism accounts for more NPF in these regions for the two boreal European mid-latitude stations". This adjustment, along with the earlier clarification in the Introduction, aims to more accurately deliver the limitations of spatial representativeness, but maintaining the relevance of the findings within the scope of the study.

- **Are the results sufficient to support the interpretations and conclusions?**

**While the major conclusion that including ammonia nucleation mechanisms at the sites investigated brings model aerosol size distributions closer to observed values in spring and summer, a number of the finer points appear either not completely supported by the data presented, or the data is presented in such a way that it is not possible to tell if it supports the conclusions reach. I will detail the problems I came across here below. While I refer to parts of the text using quotation marks, I mostly only paraphrase the text for brevity.**

**Line 226: low availability of condensable vapours postulated to explain low EC-Earth number concentrations but then shown to be not the case because ADCHEM can produce sufficient number with same condensable vapour concentrations. So why leave this in as a possible explanation? Isn't the benefit of including ADCHEM to be able to show that this is the mechanisms not the vapour concentrations that are the problem?**

The results from ADCHEM and EC-Earth3 suggest that there could be insufficient mechanisms in EC-Earth3 regarding organic gas-phase chemistry, which is better represented in ADCHEM. We suggest this could generate more condensable vapors as we see more organic aerosol mass and PNSD in ADCHEM compared to ECE3 during the summer months (shown in Fig. A4).

**Line 245: decrease in J for control case explained by decreasing ELVOC concentrations with altitude (fig 2a), which makes sense. But NH3 concentrations also decrease with altitude (fig 2c) and CLUST Js do not show the same decrease with altitude. A more nuanced discussion of the changes in J with altitude for the different mechanisms is needed here.**

Yes, this part of our results we find interesting, and the authors have included further clarifying words on Line 247: "In contrast, the CLUST nucleation rate remains unaffected by altitude. Since CLUST formation depends on temperature, ionization, and cluster scavenging sinks, higher altitudes provide more favorable conditions for nucleation in CLUST (Svenhag et al. 2024). However, this advantage is likely counterbalanced by the decreasing NH3 concentrations with altitude, which is why the formation rate remains fairly unchanged vertically."

**Fig 3 line 259: "ADCHEM better reproduces observations than EC-Earth3" − Hyytiälä ADCHEM appears to see no aerosols at all. Could this just be a plotting error?**

The authors have added a textbox in the figures where ADCHEM has this to clarify the "no-data".

**Line 265-268: It is unclear why the lower Aitken mode concentrations in the control case are ascribed to the weakness of the nucleation, but the lower Aitken mode concentrations in the CLUST cases are ascribed to limitations in how aerosols grow between modes in the model.**

This is true, we have changed the phrasing and including reference to the new figure (Fig. A8, addressed further down in our comments), this now read as:

"Nonetheless, regarding the resulting total concentration in the nucleation and Aitken mode, it agrees better with observations when using the CLUST scheme compared to using only the default ELVOC−H2SO4 nucleation. Comparing the modelled "No NPF" case with the Control case for Hyytiälä in Fig. 3 (left panel) the weak visible nucleation in the control case gives almost no growth to the Aitken mode as the two cases have the same number size distributions shown in Fig. A8 a."

The part before this paragraph is referring to the limitations in the modal structure and growth mechanic and how aerosols are moved between modes in the model for the CLUST, and the control case, (which is included when referencing: "modelled by EC-Earth3", on Line 270). This paragraph above is referring to the "total number concentration" where the weak visible nucleation in the control case" in the following sentence is ascribed to the nucleation mechanic and subsequent weak/no growth.

**Line 276: "fig 4k ELVOCs are low on April 6th Hytlemossa" – figure shows very similar ELVOC concentrations to all other days, what do the authors mean by low here?**

This was slightly unclear yes, the authors have added a reference to Fig A6 (previously Fig. A5), emphasizing the ELVOC conc. is low during this week in reference to the entire spring perspective shown in Fig. A6. Line now reads as: "ELVOC concentrations are lower during this earlier period of spring (Fig. A6)".

**Line 277: "April 6th Hytlemossa NH3 in fig 4 low compared with spring average shown in fig A5" this is very hard to see since the scale on fig A5 makes it hard to pick out where April 6th is and it requires going between two very separate figures. Suggest either marking April 6th clearly on fig A5, adding an average line on fig 4, or some other combining of the figures.**

The authors have made changes to figure A4 and figure A5 (now A5 and A6) to highlight the specific weeks. Additionally, the sentence was adjusted, and now reads as: "Additionally, ELVOC concentrations are lower at this earlier period of spring, shown in Fig. A5."

**Line 283: "H2SO4 generally higher in model level 2 than model level 1 in fig 4" – H2SO4 seems almost identical in both levels. Either a clearer figure is needed to show this is the case, or there is some error in interpretation here.**

The authors have clarified the sentence and explicitly shown with a new figure (Fig. A1): "On average, the H2SO4 concentrations are higher in the second model layer compared to the surface layer (shown in Fig A1 b)."

**Line 296: "High ELVOC-H2SO4 nucleation during the 2 week periods for both stations shown in figure 6 yield lower surface aerosol concentrations compared to NH3 nucleation as shown in figure 1" – figure 1 only shows seasonal averages, so I don't see how conclusions can be drawn by comparing nucleation rates from specific 2 week periods to number concentrations in the seasonal averages. Could the resulting number concentrations from the days shown in fig 6 not be included to make a robust conclusion here?**

The authors have added Fig. A8 showing the median PNSD of the six different studied weeks and now reads: "Despite the higher surface formation rates for ELVOC-H2SO4 nucleation during these days, this case still yields a lower total surface aerosol number concentration compared to the CLUST-High case, shown in Fig. A8 b, e."

**Line 299: "J is higher in second model layer than surface layer in fig 2" Figure 2 does not indicate where these model layers are and I suspect that both surface and 1st model layer are so close together at the bottom of this figure as to make it hard to see the changes. A clearer figure would be needed to justify this conclusion.**

This is a valid point concerning the figures, the authors have included a supplementary figure to show the surface-lower troposphere vertical distribution in Fig. A1 and added line 251 in the result section: "In Fig. A1 d the difference in gas concentrations and particle formation rates are shown at a lower altitude interval (800 hPa - 1000 hPa)." Also the mentioned Line 299 now refers to the 800-1000 hPa vertical figure (Fig. A1) instead as: "J is higher in second model layer than surface layer in fig A1"

**Line 304: why especially above 100m? Fig A2 shows J>1e-3 below 100 m also.**

The authors have removed the end of the sentence: "especially above 100 m".

**Line 308 suggests uncertainties on atmospheric concentrations of amines and iodine greater than those of ammonia, ELVOCs and H2SO4, but no sources cited to justify this claim. Considering we lack ammonia observations for most of the atmosphere, and large areas without many observations of H2SO4 and ELVOCs I'm not convinced this is the case.**

The authors have clarified this sentence and now reads as: "Studies have suggested the potential importance of amines and iodine for enhancing nucleation (Wollesen et al., 2024; Zhao et al., 2024), which are pathways not included in the EC-Earth3 model."

**Conclusion line 356 states that CLUST nucleation mechanism agrees with detailed modelling in ADCHEM; but most of the results presented show that the CLUST nucleation schemes produce less nucleation and growth than ADCHEM and the observations at the 2 sites. The conclusions here also state that CLUST nucleation mechanism agrees with results of Zhao et al. (2024), but no comparison with nucleation rates from Zhao et al (2024) was presented in the manuscript, unless I have missed something?**

We agree that many points from Zhao et al. (2024) could be explored, the authors have clarified the line by the following 2 sentences to Line 311 as: Fig. 4 in Zhao et al. (2024) also shows NH3−H2SO4 nucleation (neutral and ion-induced) as a key contributing pathway to particle formation in the upper and lower troposphere.", and Line 365 as: "Our new CLUST nucleation mechanism accounts for more NPF in this vertical region in the two northern mid-latitude stations, which has better agreement with detailed modelling in ADCHEM and NPF rates presented in Zhao et al. 2024)."

**Fig 3: ADCHEM seems to have nucleation when none is observed for Hyltemossa – this is not addressed. EC-Earth3 Hyltemossa seems to see some nucleation for all 3 observed events with all model variants, just less intense and without growth. This is not mentioned in the discussion.**

That is a good point regarding ADCHEM's performance, the authors have clarified the suggested events, and sentence now reads as: "The three events in April are captured, but underestimated in intensity and growth, with a continuously high NUS concentration throughout the day during April 6th.". The authors acknowledge there might be some bias NPF estimations in ADCHEM, it was mentioned shortly on Line 278 as: "but still show some deficiencies." A more detailed ADCHEM performance evaluation could indeed be beneficial but is not made in this study.

- **Is the description of experiments and calculations sufficiently complete and precise to allow their reproduction by fellow scientists (traceability of results)?**

**Yes**

- **Do the authors give proper credit to related work and clearly indicate their own new/original contribution?**

**Yes.**

- **Does the title clearly reflect the contents of the paper?**

**The title only partialy reflects the content of the paper. It makes no mention of the use of a complex chemistry model to improve comparison between the global model and the observations, fails to indicate that this paper is specifically addressing the role of ammonia nucleation, and refers to "boreal forests" generally, when the paper only deals with European boreal forest sites.**

- **Does the abstract provide a concise and complete summary?**

Yes, the abstract provides a very clear summary of the work.

- **Is the overall presentation well structured and clear?**

In general, the authors ask the reader to jump between figures a lot to support the scientific conclusions they are drawing, and the figures are not laid out in ways to make it easy to compare between them. This is a problem for readability and makes it hard, as a reviewer, to see if the data presented really support the conclusions drawn.

- **Is the language fluent and precise?**

**There are several small instances where use of language, or spelling or grammatical errors hinder readability of the paper. The ones I've noted are as follows:**

**Line 231: is this saying that the CLUST higher Aitken mode concentrations are due to transport either from above or from neighbouring grid boxes? This could be more clearly stated.**

The authors have clarified this paragraph, and now reads: "However, the CLUST cases still have greater Aitken-mode aerosol number concentrations at the surface due to transport. As CLUST have substantially greater aerosol formation rates in the overlying grids (or possibly in neighboring grids) and these additional aerosols can then descend (or move laterally) to the surface grid representing the station"

**Line 246 and fig A1: "CLUST schemes produce highest J in spring and autumn at high altitude" - unclear. do the authors mean that in spring and autumn at high altitude the CLUST Js are higher than control, or that in spring and autumn the high altitude Js from CLUST are higher than low altitude Js from CLUST, or that high altitude Js from CLUST are higher in spring and autumn than summer and winter? Either way it is quite hard to see this from the image plots, and without knowing what the authors classify as high altitude – UTLS? Above the BL? - perhaps time series of overage J within altitude bins might be clearer? Along with more precise text.**

The authors have clarified this paragraph with more precise wording, it now reads as: "The largest case differences here between CLUST and the control case are found at altitudes between 800-400 hPa. There are occasions when the formation rates for the Control case are occurring at altitudes between 800–400 hPa, but these rates are almost entirely produced by the

model from only BHN of water-H2SO4. During summer, nucleation throughout the troposphere seems to substantially decrease for all cases."

**Line 249-251: unclear – I'm not at all sure what is meant here**

The authors selected to remove this sentence describing the similarities to the global average, as it does not meaningfully contribute to the discussion or evaluation of the results.

**Line 277: "this variation in event strength is similar to autumn cases". It is unclear what the authors means by variation in event strength since the text is talking about a single event (April 6th).**

The authors have adjusted this Line to: "A similarly strong NPF event is captured by EC-Earth3 at Hyltemossa on the 24th of October where the resulting aerosol and H2SO4 concentrations again vary between cases (shown in Fig. A6 and Fig. A7)."

**Very minor, but noted here in-case it is helpful to the authors:**

**Line 16: particular – particulate**

Authors have adjusted this error.

**Line 162: utilized -> used (clarity of language)**

Authors adjusted "utilized" to "used".

**Line 163: spurious comma after "and NPF"**

Authors have removed this comma.

**Fig 5 caption: spelling mistake**

Authors have adjusted this error.

**Are mathematical formulae, symbols, abbreviations, and units correctly defined and used?**

**Generally yes, but there are some instances where clarity could be improved that I note here below.**

**Line 140: RICC method and DLPNO method acronyms and methods not fully explained – hard to understand the difference between the 2 datasets here**

The authors have clarified the two method versions as: "The RICC2 version method is based on Olenius et al. (2013) and the newer DLPNO version is based on Besel et al. (2020). Further details on the EC-Earth3 implemented look-up table is described in Svenhag et al. (2024).

**Line 142: do HIGH and LOW inputs mentioned here refer to CLUST-High and CLUST-low mentioned above? The change in terminology is confusing.**

The authors have clarified this in the text, now reads as: "The RICC2 (CLUST-High) version method is based on Olenius et al. (2013) and the newer DLPNO (CLUST-Low) version is based on

Besel et al. (2020b). Further details on the EC-Earth3 implemented look-up table is described in Svenhag et al. (2024). We study the ranges produced from the CLUST-High and CLUST-Low inputs, for assessing the model sensitivity to ammonia."

- **Should any parts of the paper (text, formulae, figures, tables) be clarified, reduced, combined, or eliminated?**

**There are a number of instances where either the text, figures or tables were unclear. I detail those I found here below:**

**In general the language used to refer to the different nucleation schemes could be clearer. Once I had read enough of the manuscript I understood that base = binary + Riccobono, CLUST = binary + Nh3 ternary, and then Clust+Riccobono is binary + NH3 ternary + Riccobono, but this could be made clearer in table 1, and earlier in the text.**

**Table 1: fuller description of the difference between the nucleation schemes would be helpful here. It is hard as a reader who isn't intimately familiar with Svenhag 2024 to understand from the text and this table the differences between the different model runs**

This is a very valid point, we have now extended the information described within Table 1 to include every involved NPF mechanism and the corresponding species.

**Fig 1: very difficult to see what is going on in nucleation mode because of dominance of accumulation mode. Can't see ADCHEM line for Hyytiälä except for in summer.**

Yes, it is unfortunate that some parts of nucleation and accumulation mode are not visible. But using a log-scale would remove the visible differences in the Aitken mode, which have the highest median divergence between the model cases. For ADCHEM not being visible at Hyytiälä except in summer: we give this description in the caption: "Note, the ADCHEM model only generated output from May to August for Hyytiälä and thus is only shown for Jun-Aug."

**Table A1 and line 233-237: The CRE change from CLUST High in summer for Hyltemossa is also very large and not mentioned, and strange that for the same period and simulation the change for Hyytiälä is 0. The authors are correct to point out the problem of looking only locally at CRE and DRE changes when advection and transport play such a role. The very limited discussion presented here however raises more questions than it answers. Either a more thorough discussion is warranted, or the CRE and DRE discussion should be omitted from this paper altogether (another paper by the same author is referenced – so perhaps it is not necessary to include that analysis here also).**

After consideration, the authors also agree with this point. This table was intended to be displayed in comparison to Svenhag et al. (2024) and does not directly contribute to the discussion of what is studied in this paper. We have decided to omit this table and the short paragraph description in the result section.

**Line 240 and fig 2: ELVOC-H2SO4 and NH3-H2SO4 don't directly relate to how the mechanisms are named in the legend, so harder for the reader to quickly relate text to plot. It would help to remind that the first is called "control" and the second is both of the "CLUST" cases. It would help to relate the second model layer to a pressure since that is the vertical axis of the graph. Or even to put a line on the graph indicating where the 2nd**

**model layer starts. Also it is not clear from the figure that the CLUST cases and control case are similar at altitudes below 100m, which is implied in the text.**

These are good points. The referenced height and model cases were indeed not clear in context with the figure, so the authors changed this line to read as: "(approx. at 900 hPa and above) when comparing the control case (ELVOC-H2SO4) and CLUST cases (NH3-H2SO4)". We have addressed the "invisible" 2nd model layer by adding another figure where we only show (zoomed in) 1000-800 hPa vertical distribution of Figure 2 with each layer is assigned a marker.

**Fig 3 and 4: would be helpful to have these on same x-axis for direct comparison**

The authors agree with this and have merged figures of the weekly studied cases. Now Fig 3 and 4 is one larger figure, similarly for the summer and autumn case-figures.

**Fig 4: what are the dashed lines in e,k,f,l? also e and k show hints of lines in multiple colors but the legend says these are only the no NPF case, so then what are the other colors for?**

Thank you for noting this, In this caption the: "The ELVOC and NH3 concentrations are from the No-NPF case", is not true and is removed, and we further added an explanation in the caption to what the dashed lines indicate as: "The concentrations for ELVOC and NH3 level 2 are shown as dotted lines in e, f, k, l."

- **Are the number and quality of references appropriate?**

**In general, yes.**

**There was one instance in the introduction where the availability of vertical profiles of aerosol properties seemed to be understated and the relevant references missed. This is at line 70-75, which mentions a "lack of vertical profiles" and "aerial campaign measurements are not yet sufficient" for sub-100nm aerosol observations. While I believe there may not be vertical profiles made directly over both stations considered here, some airborne observations have been made over Hyytiälä and campaigns over other regions have made a substantial number of vertical profiles from aircraft e.g. such as ATom, Café-Brazil etc. The state of the field would be better represented if reference were made to these, although they are may not help directly with the analysis presented here.**

The authors agree with this point, we have included a short mention of the availability of aerial campaigns on Line 71. As: "Some aeronautical measurement campaigns from e.g. ATom (Brock et al., 2019) and CAFE-Brazil (Curtis et al., 2024) at various locations have made efforts to capture this, but they are not evaluated in this study." With the following 2 references:

*Brock, C. A., Williamson, C., Kupc, A., Froyd, K. D., Erdesz, F., Wagner, N., Richardson, M., Schwarz, J. P., Gao, R.-S., Katich, J. M., Campuzano-Jost, P., Nault, B. A., Schroder, J. C., Jimenez, J. L., Weinzierl, B., Dollner, M., Bui, T., and Murphy, D. M.: Aerosol size distributions during the Atmospheric Tomography Mission (ATom): methods, uncertainties, and data products, Atmospheric Measurement Techniques, 12, 3081–3099, https://doi.org/10.5194/amt-12-3081-2019, 2019.*

*Curtius, J., Heinritzi, M., Beck, L. J., Pöhlker, M. L., Tripathi, N., Krumm, B. E., Holzbeck, P., Nussbaumer, C. M., Hernández Pardo, L., Klimach, T., Barmpounis, K., Andersen, S. T., Bardakov, R., Bohn, B., Cecchini, M. A., Chaboureau, J.-P., Dauhut, T., Dienhart, D., Dörich, R., Edtbauer, A., Giez, A., Hartmann, A., Holanda, B. A., Joppe, P., Kaiser, K., Keber, T., Klebach, H., Krüger, O. O., Kürten, A., Mallaun, C., Marno, D., Martinez, M., Monteiro, C., Nelson, C., Ort, L., Raj, S. S.,*

*Richter, S., Ringsdorf, A., Rocha, F., Simon, M., Sreekumar, S., Tsokankunku, A., Unfer, G. R., Valenti, I. D., Wang, N., Zahn, A., Zauner-Wieczorek, M., Albrecht, R. I., Andreae, M. O., Artaxo, P., Crowley, J. N., Fischer, H., Harder, H., Herdies, D. L., Machado, L. A. T., Pöhlker, C., Pöschl, U., Possner, A., Pozzer, A., Schneider, J., Williams, J., and Lelieveld, J.: Isoprene nitrates drive new particle formation in Amazon's upper troposphere, Nature, 636, 124–130, https://doi.org/10.1038/s41586-024-08192-4, 2024.*

- **Is the amount and quality of supplementary material appropriate?**

**The reader is asked frequently to refer to supplementary figures to justify major conclusions in the main text. It might improve readability if more of the relevant parts from supplementary figures were combined with the relevant figures in the main text.**

**I have mentioned above a number of areas where conclusions drawn do not seem to be fully supported by the data presented. It may therefore be necessary to include further supplementary material, but in my opinion, the paper might instead benefit from changes to the main figures presented and a re-evaluation of some of the conclusions drawn.**

We thank the reviewer for all this valuable feedback. We appreciate the observations that some conclusions may not have been fully supported by the data as originally presented. In response, we have re-evaluated the relevant sections of the manuscript from all the above comments and points and have taken the following steps to strengthen the support for our conclusions.

---

## Author Comment (AC3)

**Author Response RC1**

We thank the reviewers for their thorough and constructive comments, which have helped to improve the quality and clarity of our manuscript. We have carefully considered all suggestions and made corresponding revisions in the manuscript. Below, we provide detailed responses to each comment, with changes clearly indicated in the revised version of the manuscript.

Reviewer comments are reproduced in **bold**, and our responses follow each comment in **plain text**. Line numbers refer to the revised manuscript unless otherwise stated.

- Start of RC1 Comments: -

**In a previous paper, Svenhag et al (GMD 2024) implemented a mechanism for new particle formation (NPF) from H2SO4 and NH3 in EC-Earth and found minor differences in CCN, but despite this they found a change to the radiative effect of clouds of 0.28-1Wm-2. They performed a model evaluation, checking the aerosol size distribution at 12 surface stations worldwide.**

**In this paper, the authors focus on simulating NPF events at two surface stations using (I think) the same EC-Earth simulations, and compare them to an ADCHEM model run, presumably one of those published in de Jonge et al (E, S & T 2024). A new EC-Earth simulation without NPF is included, but only features in part of the analysis.**

**The simulations overestimate primary emissions in winter. ADCHEM has some different NPF mechanisms including iodine and dimethylamine, and no organic NPF, and performs better.**

**There's an interesting set of plots comparing simulated and observed size distributions and "bananas" at two sites. However, given that most of the simulations (as far as I can tell) are already published and a very similar evaluation of ADCHEM at these sites was already published by de Jonge et al (2024) e.g. their Figure 3, it left me wondering whether the paper currently satisfies the "substantial new concepts, ideas, methods, or data" review criterion for ACP. I think it could, if the analysis were broadened and deepened to probe the model more comprehensively and replace some of the speculations in Section 3.2 with additional detailed analysis or sensitivity studies. This is likely to need new simulations.**

**Major comments**

**In my assessment, the paper relies for its novelty on two aspects**

1. **The evaluation of simulated size distributions as a function of time ("banana plots") in EC-Earth, which is interesting and not often done with aerosol-climate models.**

2. **The comparison between ADCHEM and EC-Earth.**

**One could argue that the value of (1) is limited for people outside the EC-Earth community because the size distributions are dominated by the effects of the coupling to the IFS meteorology, and it's hard to disentangle the odd behaviour resulting from this this from the effects of the NPF. However, it's still interesting to see.**

**The value of (2) is currently limited since the authors only show one ADCHEM simulation with very different nucleation mechanisms to the EC-Earth simulations. Perhaps there is an opportunity here to analyse the other ADCHEM simulations published by de Jonge et al,**

**which include simulations that only include H2SO4-NH3 NPF that would (at least to an outsider) seem to be a fairer comparison to the EC-Earth CLUST simulations.**

**If the sentence in the abstract "When comparing diurnal EC-Earth model results with ADCHEM and observations, we establish that using solely organic-H2SO4 nucleation parameterization will underestimate the aerosol number concentrations" is not to be misleading, the authors should test what happens when ELVOC nucleation from Riccobono et al (2016) is included in ADCHEM. Currently ADCHEM does not tell us about the organic-H2SO4 nucleation parameterization.**

We thank the reviewer for their insightful comments and appreciate the opportunity to clarify our work. Our primary aim with this study is to present a detailed characterization of the EC-Earth3 model's aerosol size distribution, particularly highlighting the diurnal, seasonal, and vertical variability introduced by the newly implemented nucleation mechanism. Previous evaluations have largely relied on long-term averages, potentially masking important sub-daily and seasonal dynamics that are crucial for understanding the performance and behavior of aerosol-climate models. Our results indicate that performing detailed comparisons between Earth system models and observations at a high temporal resolution reveal previously unexplored insights into model structures. We believe that this process-level focus, e.g. nucleation events, adds unique information to existing model evaluation efforts, even though the results may be especially relevant to those working within the EC-Earth community.

Concerning the comparison to ADCHEM, we fully agree that such comparisons can be powerful tools for identifying model behavior and constraints. In this work, our intention was to use ADCHEM as a reference for diurnal variability, as well as to validate the general trends in particle number concentrations and growth. However, we acknowledge the reviewer's point that the comparison is limited by the fact that the ADCHEM simulation shown uses different nucleation mechanisms from the EC-Earth3 CLUST configuration. We agree that additional ADCHEM simulations (including those with simpler nucleation mechanisms e.g., $H_2SO_4$-$NH_3$ only) could provide more model insight.

Nevertheless, performing such comprehensive work of additional ADCHEM simulations, including detailed sensitivity experiments, would extend beyond the scope of this paper. The primary reason for this is that ADCHEM and EC-Earth3 are fundamentally different in terms of model architecture, spatial resolution, and process representation. So, changes in nucleation parameterizations in ADCHEM are unlikely to yield results that are directly analogous to modifications in EC-Earth3, limiting the interpretability of such a comparison within the context of this study.

With that in mind, our study aims to assess differences between a coarser Earth system model and a detailed process model rather than to reproduce identical nucleation conditions in both models. The ADCHEM simulation included in our study was chosen to highlight the contrast in nucleation parameterizations and model frameworks. In response, we have revised parts of the text and added new figures, which include further analysis, seen in the new manuscript version.

**Minor comments**

–**In Figure 2e, NPF is so non-linear that the annual mean NPF rate must be dominated by values close to zero. Would something like the 95th percentile make more sense instead?**

That is an interesting point. After testing this, the authors have adjusted and analyzed this plot using the 95th percentile, this elevates the disparity between the CLUST and Control case in the free troposphere further. The plot now shows the 95th percentile, including the added Fig. A1 of 1000-800 hPa.

–**Using the Jokinen et al (2015) yields for ELVOC with the Riccobono et al 2014 nucleation mechanism likely leads to high uncertainty. More discussion of this might be useful in the light of developments to organic aerosol schemes in ADCHEM by the paper's coauthors (e.g Roldin et al, Nat Comms 2019).**

The authors have now addressed this in the Further discussion section on Line 327 as: "Additional uncertainties in the EC-Earth3 aerosol chemistry can also arise from the simplifications in modeling organic oxidation yields (from Jokinen et al. 2015) and growth processes. However, incorporating more complex chemical mechanisms is often constrained by the computational costs associated with most Earth system models."

–**A table of the NPF mechanisms in ADCHEM, and in general a more detailed description of how ADCHEM simulates particle formation and growth and how fair the comparison with a climate model is, would be useful.**

While this is a valid point, adding further ADCHEM description does not contribute much to our paper's focus here, the authors however, would refer the reader to our references Wollesen de Jonge et al. (2024) and Roldin et al. (2019) where the model mechanisms are described in detail.

–**The differences between the CLUST schemes are described, but the value of including two EC-Earth simulations with different CLUST NPF mechanisms would be greater if differences  (or similarities) between the behaviour of the simulations (ie the number of particles formed) with these schemes were discussed in the text.**

Good point, the authors do acknowledge that a more detailed discussion on the specific behavior of the simulations, such as the number of particles formed, could provide valuable insight. Svenhag et al. (2024) focuses on a deeper analysis of the CLUST-High and CLUST-Low differences, where we study the behavior of these two schemes in more detail.

–**I liked Figure 3 in principle, but it could be tidied up a bit with larger labels, especially the colorbar. Also just reading the caption I was a bit confused why ADCHEM had no aerosols at Hyytiala but did have the data at Hyltemossa (maybe just add to the caption ", while for Hyltemossa the simulation was run from to ").**

We appreciate these beneficial suggestions; the authors have adjusted Fig. 4 to account for the ADCHEM with a box saying: "No Data". Additionally, larger labels have been added to these figures showing weekly trends.

–**Figure 4 also needs larger labels.**

The Authors have adjusted Fig. 4 and made the labels larger.

–**The sudden discontinunities in the aerosol size distribution around midday each day are attributed to the coupling of the chemical transport model to IFS. This is interesting. Because NPF is nonlinear, you would expect a systematic bias to result. Can this be studied further? How hard would it be to increase the frequency of the coupling?**

We agree that this coupling introduces potential for systematic biases, particularly given the nonlinear nature of NPF processes. On one hand, It would indeed be interesting to conduct a further study focused on the impact of this coupling error on the simulation results, especially considering the variability in particle formation that could result from the temporal resolution of the coupling. However, we would like to note that future model versions of EC-Earth (currently under development) in OpenIFS48r1, the coupling between the chemistry and meteorology will occur within in every time step (30-60 min), which is expected to significantly reduce the error introduced by this temporal discrepancy.

–**Line 315 and elsewhere: sentences could be improved, a number of typos to fix.**

The authors have made changes to multiple sentences and sections, including Line 315.

---

## Author Response (AR2)

**Editor Revisions: Author Response to Peer Reviewer Report 1 from RC3**

The authors would like to express their gratitude to the editor and reviewers for their thoughtful and constructive feedback throughout the review process. Their comments and suggestions have significantly contributed to improving the clarity and quality of this manuscript.

Reviewer comments are reproduced in **bold**, and our responses follow each comment in **plain text**. Text from the previous Author response is produced in **red**. Line numbers refer to the revised manuscript unless otherwise stated.

- Start of Comments -

**- The basic 10 subitems of comment 4**

Comment: **4. Are the scientific methods and assumptions valid and clearly outlined? To some extent. Some parts of the methods section are not clear to me:**

**- line 112: what are "specific dimensions"?**

The authors addressed this comment in the last round of reviews (Author Response RC3), and adjustments were made accordingly. The response read as: The authors see that this was incorrectly described, where we were intending to refer to the size and log-normal structure of M7 here for the modal bins. This has been corrected and now reads on line 115: "However, this modal system in M7 limits the size distribution appearance to fall within specific log-normal modes at fixed sizes, ..."

**- line 127: what does "explicitly modelled" mean?**

The authors recognize the unclear language here of "explicitly modelled" and have made changes to this description of the model process in the text. This line 115 now reads as:

"The KK function in M7 determines the particle formation rate using available gas phase ELVOCs and H2SO4 concentrations for estimated particle survival through condensational growth. Only after this growth to 5 nm are the aerosols introduced into the modal size distribution (Bergman et al., 2022)."

Previously this line read as:

*"The KK function in M7 determines the particle formation rate using available gas phase ELVOCs and H2SO4 concentrations for estimated particle survival through condensational growth, and the resulting particles then enter the explicitly modelled size distribution in the nucleation mode (Bergman et al., 2022)."*

**- line 138: how are the five dimensions of the lookup table discretized?**

The authors have addressed this further by adding the following description for the lookup table clarifying what was brought up in this comment. Following Line 139 now reads as:

"The lookup table contains nucleation rates for all combinations of the five variables that define the rate, and rates at specific conditions are determined by multivariate interpolation. This approach ensures accurate rates, avoiding the typical problem of nucleation rate parameterizations, namely that the rate may not be reproduced under all different conditions of "the parameter space"."

**- line 139: how is the "cluster scavenging sink" computed?**

The authors addressed this comment in the last round of reviews (Author Response RC3), and adjustments were made accordingly. The response read as: The authors have added this as a clarifying Line 143: "The (5) molecular cluster scavenging sink in CLUST is calculated from sulfuric acid condensation sink which is scaled for different cluster sizes (Yazgi and Olenius,2023b; Lehtinen et al., 2007). In the present EC-Earth3 setup, the input total condensation sink of sulfuric acid to CLUST is calculated from all 7 aerosol modes at every model time step."

**- line 142: what are the dimensions of the IPR lookup table?**

Authors have included a further description of the IPR table in the Methods section, on line 154:

"An IPR lookup table with global coverage of galactic cosmic rays and soil radon is used (Yu et al., 2019). It reads model pressure (203 altitude levels), magnetic latitude (91 bands), and the model grid land cover fraction (for $^{222}$Rn) to generate the ion production rates, analogous to the implementation in Svenhag et al. (2024)."

**- line 169: there seem to be different emissions for EC-Earth3 and ADCHEM. how does that influence the results?**

The authors have included a further discussion addressing this point made by the reviewer. Adding onto the discussion between the models, line 343 now reads:

"There are also some differences in the emission inventories used by ADCHEM and EC-Earth3. However, these discrepancies are unlikely to be the primary source of emission-related differences observed at the two stations. A more significant factor is the spatial resolution at which each model reads the emissions. EC-Earth3 uses areal mean emissions from the CMIP6 inventory, interpolated over a coarse $2° \times 3°$ (longitude $\times$ latitude) grid. In contrast, ADCHEM utilizes the CAMS inventory at a much finer resolution of $0.1° \times 0.1°$ extracting emission values directly from individual grid cells."

**- line 167: how is the trajectory discretized in time? how is the column discretized in space?**

The authors recognized that clearer model description of ADCHEM was necessary here, and these following lines have been adjusted to clarify in the Method section (2.4):

Line 179: "Back-trajectories were simulated seven days backwards in time using the HYSPLIT default output interval of 1 hour for the Hyytiälä and Hyltemossa field stations. The trajectory coordinates were then linearly interpolated to a temporal resolution of 10 minutes when calculating the emissions of gases and particles from the urban areas, ocean, and forested regions surrounding the stations."

Line 181: "In this version of ADCHEM, the one-dimensional column model consists of 20 vertical layers spaced logarithmically, starting from the first layer that representing the lowest 10 meters up to the 20th layer which represents the atmospheric layer between 1900 m and 2100 m above ground level."

Line 186: "ADCHEM used a main model time step of 60 seconds when solving the atmospheric chemistry, aerosol dynamics and vertical mixing."

**- line 181: "Particles and gasses were mixed by use of the GDAS" how does that work in ADCHEM?**

The authors have again included a further description of the ADCHEM model to answer the "**-line 183 & -line 181:**" questions:

Now, a longer paragraph on line 198 reads as:

"For the gases, the model simulates the gas-phase and aqueous-phase chemistry of 5005 species via 13062 reactions. Strong inorganic acids ($H_2SO_4$, $HNO_3$, $HCl$, $HIO_3$), ammonia and organic oxidation products with a pure liquid saturation vapour pressure less than $10^{-2}$ Pa at 293 K (in total 873 species) were treated as potentially condensable vapours and represents the particle size dependent condensation and evaporation dynamics. Other water-soluble gases such as $SO_2$ and the DMS oxidation product MSIA are further oxidized in the aerosol and cloud droplet aqueous phase, forming lower volatility products like sulfate and MSA that likewise help to growth the particles.

The model solves the atmospheric diffusion equation in the vertical direction. The vertical diffusion coefficients ($K_z$) were calculated based on a slightly modified Grisogono scheme (Jericevic et al., 2010; Öström et al., 2017), where the $K_z$ depends on the height above ground, the friction velocity and the height of the atmospheric boundary layer, which is sourced from the GDAS meteorology. "

And on line 205:

"A more detailed description of the model along with specific cases where the model has been used can be found in Roldin et al. (2019) and Wollesen de Jonge et al. (2024)."

These added lines also include two added references in the main text:

Öström, E., Putian, Z., Schurgers, G., Mishurov, M., Kivekäs, N., Lihavainen, H., Ehn, M., Rissanen, M. P., Kurtén, T., Boy, M., Swietlicki, E., and Roldin, P.: Modeling the role of highly oxidized multifunctional organic molecules for the growth of new particles over the boreal forest region, Atmos. Chem. Phys., 17, 8887–8901, https://doi.org/10.5194/acp-17-8887-2017, 2017.

Jericevic, A., Kraljevic, L., Grisogono, B., Fagerli, H., and Vecenaj, Ž.: Parameterization of vertical diffusion and the atmospheric boundary layer height determination in the EMEP model, Atmos. Chem. Phys., 10, 341–364, https://doi.org/10.5194/acp-10-341-2010, 2010.

**- line 183: "The ADCHEM model thereby attempts to reproduce the concentration of gasses and particles" how does that work in ADCHEM?**

See text above and the provided additional references to previous ADCHEM publications.

**- line 231: what grids are used in EC-Earth3?**

      Authors here repeat the previous comment from our Author Response to AC3, pointing to what is already described in the paper, in method section, line 101:

" The IFS model time step is 45 minutes and set to generate output every 3 hours on 100 a 0.7°spectral truncation grid. TM5 uses hourly time steps and is set to produce hourly model output with 2° × 3° (latitude× longitude) resolution. The vertical resolution in TM5 is represented by 34 hybrid sigma pressure levels, and IFS have the same hybrid pressure levels, but extrapolated to 91 layers."

**11. Is the language fluent and precise?**
**Mostly yes. Some sentences are rather long and complex and should be rephrased for a better readability.**

      Following both the initial and the current round of the review process, we examined the manuscript for instances of long or complex sentence structures highlighted by the reviewers. Where appropriate, we have rephrased and simplified these sentences to clarify and improve overall readability.

**12. Are mathematical formulae, symbols, abbreviations, and units correctly defined and used?**
**Mostly yes. Mathematical symbols sometimes appear in text font instead of math font. For example, particle diameter d on lines 111 and 112 or nucleation rate J on line 124. The long terms on lines 176-178 should be simplified.**

The authors have adjusted these text fonts on line 124, 111 and 112 to mathematical font accordingly.
The authors recognize these terms are long. However, the terms: "*DLPNO-CCSD(T)/aug-cc-pVTZ//ωB97X-D/6-31++G(d,p)*", "*DLPNO-CCSD(T)//M06-2X*", and "*RI-CC2//ωB97X-D*" are description terms decided by model development to signify specific versions and enable simpler reproducibility and consistency.

**13. Should any parts of the paper (text, formulae, figures, tables) be clarified, reduced, combined, or eliminated?**
**Some figures should be revised. There are no subplot labels in figures 3, 5, A1, and A6. The font sizes in figures 3, 4, 5, 6, A4, A5, A6, and A7 are too small. The captions of some figures are not self-explanatory. In figures 3 and A6, it is unclear why the ADCHEM base case has a non-zero value of 10 cm-3. In figures 4, 6, and A7, its is unclear what "modelled layer 1 and 2" or "level 1" and "level 2" mean.**

Following both the initial round of review revisions, figures have been adjusted, combined and clarified. The following are the main changes for the figures referenced above:

- The non-zero value for ADCHEM in Figure 3 has been addressed more clearly both in the method section and in the figure caption. The caption for Figure 3 and 4 (combined) now reads as: "**Figure 3.** The top two figures show the surface aerosol number size distribution over the springtime for the 5 modelled EC-Earth3 cases, with the DMPS measured aerosols at Hyytiälä and Hyltemossa. ADCHEM simulations have no available hourly data outside the Summer months for Hyytiälä. The bottom section shows EC-Earth3 modelled layer 1 and layer 2 Hyytiälä and Hyltemossa springtime cases showing (a, b, g, h) daily maximum particle formation rate, (c, d, i, j) H2SO4, (e, k) ELVOC, (f, l) and NH3 gas concentration. The concentrations for ELVOC and NH3 level 2 are shown as dotted lines in e, f, k, l. The missing particle formation rates below 10-4 in panel a, b, g, and h are considered practically zero."
- On line 272 the terms "model level 1" and "model level 2" are described.
- The font sizes in Figures 3, 4, 5, 6, A4, A5, A6, and A7 have been slightly adjusted to avoid obscuring any information. We remain fully open to making further refinements prior to final submission, if needed.

**Editor Revisions: Author Response to Peer Reviewer Report 2 from RC1**

The authors would like to express their gratitude to the editor and reviewers for their thoughtful and constructive feedback throughout the review process. Their comments and suggestions have significantly contributed to improving the clarity and quality of this manuscript.

Reviewer comments are reproduced in **bold**, and our responses follow each comment in **plain text**. Line numbers refer to the revised manuscript unless otherwise stated.

- Start of Comments -

**Minor comments**
**I previously wrote: 'If the sentence in the abstract "When comparing diurnal EC-Earth model results with ADCHEM and observations, we establish that using solely organic-H2SO4 nucleation parameterization will underestimate the aerosol number concentrations" is not to be misleading, the authors should test what happens when ELVOC nucleation from Riccobono et al (2016) is included in ADCHEM. Currently ADCHEM does not tell us about the organic-H2SO4 nucleation parameterization.'**
**I still think the abstract should be revised so that the sentence doesn't imply that the ADCHEM simulation provides insights into the whether or not the organic-H2SO4 nucleation parameterization can explain aerosol number concentrations (even though I accept that the observations could provide these insights).**

The authors agree with the concern and have revised the sentence in the abstract to avoid implying that the ADCHEM simulation provides insights into the organic–$H_2SO_4$ nucleation parameterization. The updated wording now more accurately reflects that the insights are based on comparison with observations, as suggested. The referenced line 10 in the abstract now reads as:

"When comparing diurnal EC-Earth3 model results with in-situ observations at an hourly temporal resolution, we establish that using solely organic–H2SO4 nucleation parameterization will underestimate the aerosol number concentrations. The new added NH3–H2SO4 nucleation parameterization in this study improves the resulting aerosol number concentrations and reproduction of particle formation events with EC-Earth3. However, from March to October, the EC-Earth3 still underestimates particle formation and growth."

**L255: "unaffected by altitude" – I think this would be clearer if it were qualified, for example "approximately unaffected by altitude due to compensating effects"**

The authors agree and have altered this sentence accordingly, line 263 (previously L255) now reads as:
"In contrast, the CLUST nucleation rate remains approximately unaffected by altitude because of compensating effects, as its formation is governed by temperature, ionization, and cluster scavenging sinks."

**L344: "Assuming the…" sentence can be improved with more scientific language**

The authors regognized the unclear language here and have revised the sentence, Line 339 now reads as:
"The inclusion of some detailed chemical processes, as represented in the ADCHEM model, may yield results of sufficient relevance to climate impacts to justify the associated increase in computational expense when coupled with EC-Earth3."

**L372: sentence starting "It" can be improved – perhaps "It" -> "This artefact" and "aerosols…influence" or "aerosol..influences"**

The authors have adjusted this sentence accordingly. Line now reads:
"This artifact may also introduce uncertainty regarding the influence of near-surface aerosols on cloud–aerosol interactions in EC-Earth3 within the 6-hour coupling window."

**L385: "suggests"->"suggest"**
This word has been adjusted accordingly.

**L387: "more NPF"- specify what "more" is relative to**
The authors see this language error, and this word have been adjusted to "higher NPF rates" which should be the correct term here.

**L390: "the inclusion"->"should be included"**
This sentence was adjusted in the **L344** comment above.

**L394 "potential false representation from" phrasing could be improved**

This sentence has been adjusted to clarify the conclusion, line now reads:
"This study further underscores the potential for misrepresentation when relying solely on annual median values to evaluate the performance of EC-Earth3 (and other ESMs), due to pronounced seasonal variability."

**L397: would be good to add a reference to support the assertion that the climate forcing from aerosols has large seasonal variation**
For this statement, the authors have now included the reference to support this, the two (also previously cited) sources to corroborate this claim on this line:
"Forster et al. (2021)"(CMIP6 report, Chapter 7) and "Carslaw et al. (2013)".

---

## Author Response (AR3)

**Editor Revisions: technical corrections**

The authors again give thanks to the editor for their constructive feedback throughout the review process.

Reviewer comments are reproduced in **bold**, and our responses follow each comment in **plain text**. Line numbers refer to the revised manuscript unless otherwise stated.

- Start of Comments -

**Thank you for the additional effort on revisions in response to referee comments. Note incomplete sentence at line 200.**

The authors have corrected this error in the pdf document, and the full sentence now reads on line 200:

"Strong inorganic acids ($H_2SO_4$, $HNO_3$, $HCl$, $HIO_3$), ammonia and organic oxidation products with a pure liquid saturation vapour pressure less than $10^{-2}$ Pa at 293 K (in total 873 species) were treated as potentially condensable vapours and represents the particle size dependent condensation and evaporation dynamics. Other water-soluble gases such as $SO_2$ and the DMS oxidetion product MSIA are further oxidized in the aerosol and cloud droplet aqueous phase, forming lower volatility products like sulfate and MSA that likewise help to grow the particles."